# Naked mole-rats have distinctive cardiometabolic and genetic adaptations to their underground low-oxygen lifestyles

Chris G. Faulkes [1,8], Thomas R. Eykyn [2,8], Jan Lj. Miljkovic [3,8], James D. Gilbert[1], Rebecca L. Charles[4], Hiran A. Prag [3,5], Nikayla Patel [6], Daniel W. Hart[7], Michael P. Murphy [3], Nigel C. Bennett[7] & Dunja Aksentijevic [6] ✉

The naked mole-rat *Heterocephalus glaber* is a eusocial mammal exhibiting extreme longevity (37-year lifespan), extraordinary resistance to hypoxia and absence of cardiovascular disease. To identify the mechanisms behind these exceptional traits, metabolomics and RNAseq of cardiac tissue from naked mole-rats was compared to other African mole-rat genera (Cape, Cape dune, Common, Natal, Mahali, Highveld and Damaraland mole-rats) and evolutionarily divergent mammals (Hottentot golden mole and C57/BL6 mouse). We identify metabolic and genetic adaptations unique to naked mole-rats including elevated glycogen, thus enabling glycolytic ATP generation during cardiac ischemia. Elevated normoxic expression of HIF-1α is observed while downstream hypoxia responsive-genes are down-regulated, suggesting adaptation to low oxygen environments. Naked mole-rat hearts show reduced succinate levels during ischemia compared to C57/BL6 mouse and negligible tissue damage following ischemia-reperfusion injury. These evolutionary traits reflect adaptation to a unique hypoxic and eusocial lifestyle that collectively may contribute to their longevity and health span.

Naked mole rats (*Heterocephalus glaber*, NMRs) are characterised by their social insect-like (eusocial) behaviour defined by a reproductive division of labour, overlapping generations, and cooperative care of the young. They exhibit a suite of adaptations to living in a hypoxic (low oxygen, $O_2$) subterranean environment[1]. The NMR (Fig. 1A) is one species within a clade of >30 species of African mole-rats, all of which adopt a subterranean lifestyle across sub-Saharan Africa within differing soil types and degrees of sociality (from solitary to eusocial,

Fig. 1B, C)[2–7]. Cooperative breeding and eusociality have evolved convergently within the African mole-rat clade. However, the NMR's ancestral lineage diverged from their common ancestor ~30 million years ago[8]. Despite similarities in their subterranean lifestyle, it is accepted that NMRs have many unique aspects to their biology compared to other African mole-rat species[1].

Up to 300 NMRs live together in an extensive complex composed of labyrinths of underground tunnels, toilet chambers, and a

[1]School of Biological and Behavioural Sciences, Fogg Building, Mile End Road, Queen Mary University of London, London E1 4NS, UK. [2]Department of Imaging Chemistry and Biology, School of Biomedical Engineering and Imaging Sciences, King's College London, St Thomas' Hospital, London SE1 7EH, UK. [3]MRC Mitochondrial Biology Unit, University of Cambridge, Keith Peters Building, Cambridge CB2 0XY, UK. [4]Centre for Clinical Pharmacology and Precision Medicine, William Harvey Research Institute, Bart's and the London Faculty of Medicine and Dentistry, Queen Mary University of London, London EC1M 6BQ, UK. [5]Department of Medicine, University of Cambridge, Cambridge Biomedical Campus, Cambridge CB2 0QQ, UK. [6]Centre for Biochemical Pharmacology, William Harvey Research Institute, Bart's and the London Faculty of Medicine and Dentistry, Queen Mary University of London, London EC1M 6BQ, UK. [7]Department of Zoology and Entomology, Mammal Research Institute, University of Pretoria, Pretoria 0002, South Africa. [8]These authors contributed equally: Chris G. Faulkes, Thomas R. Eykyn, Jan Lj. Miljkovic. ✉e-mail: d.aksentijevic@qmul.ac.uk

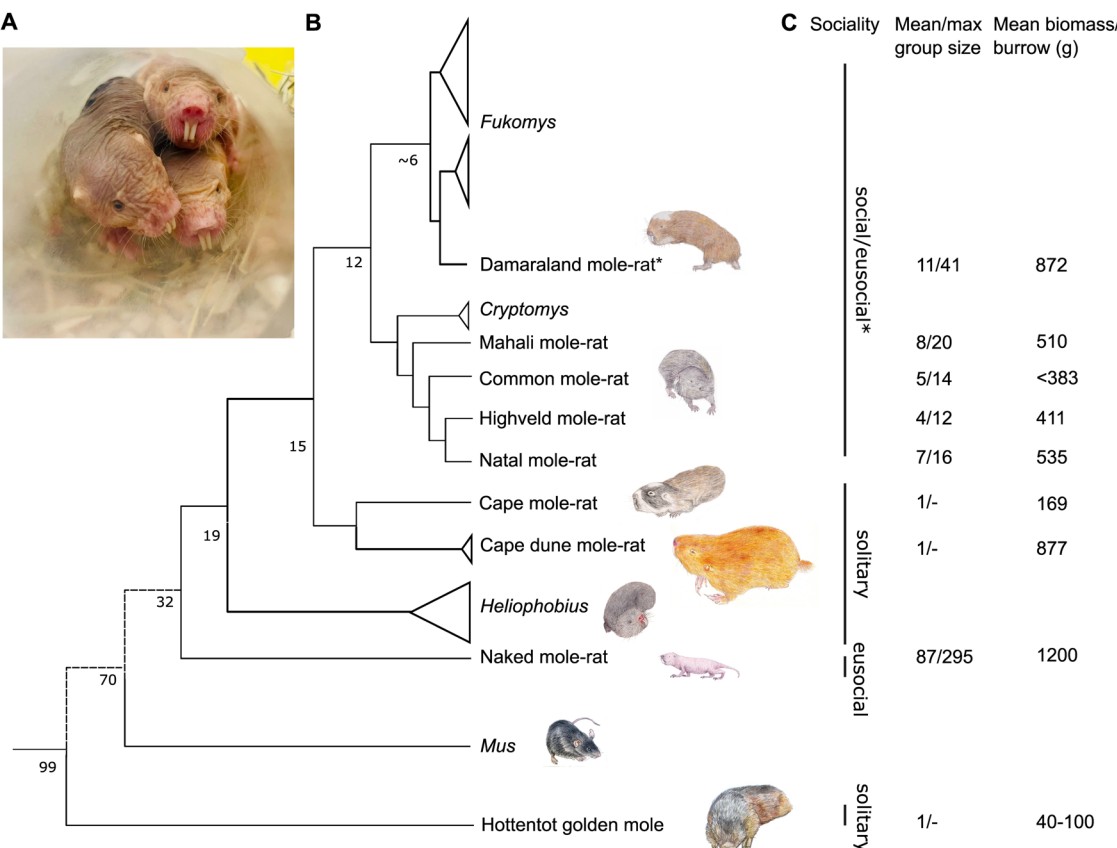

**Fig. 1 | Phylogenetic relationships and social status of African mole-rats, the mouse and the hottentot golden mole. A** Naked mole-rats (*Heterocephalus glaber*) from the captive colonies at Queen Mary University of London. **B** Simplified molecular phylogeny for the African mole-rats indicating main clades/genera and species sampled in this study, together with the mouse *Mus* and the hottentot golden mole, *Amblysomus hottentotus*. Mole-rat phylogenetic relationships are based on mitochondrial 12 S rRNA and cytochrome-b sequence data, and analysis of 3999 nuclear genes. Numbers on internal nodes represent approximate divergence times in millions of years ago (myr)[10]. **C** Social lifestyles, mean and maximum social groups sizes, and mean biomass of animals per burrow as indicated[2-7].

communal nest dug in fine, hard-packed soil, foraging for their staple diet of roots and tubers. Characteristically, mole-rats huddle together in nest chambers that are often many metres distant from transient molehills, which open to the surface and can be up to 0.9 m deep[9]. In this environment oxygen ($O_2$) regularly becomes scarce and carbon dioxide ($CO_2$) levels become elevated[10,11]. Out of all the species of African mole-rats, the potential for hypoxia as well as $CO_2$ accumulation is greatest in NMRs as they have the longest cumulative burrow length (up to 3-4 km total tunnel length), the largest absolute group sizes and colony biomass that exceeds other mole-rat species (Fig. 1C)[2-7]. Collectively, these living conditions limit NMRs gaseous exchange with the surface world. While data on $O_2/CO_2$ levels in wild NMR burrows is limited[11] and has never been measured in a nest chamber full of animals, NMRs in captive colonies are able to tolerate hours of extreme hypoxia (5% $O_2$ for up to 300 minutes), and can even survive up to 18 minutes of anoxia[12]. NMRs often elect to spend more time in areas of their burrow system with extreme atmospheric conditions including the nest chamber, where they may spend up to 70% of their time[13]. This is something not regularly seen in other social African mole-rat species.

This challenging hypoxic habitat creates strong selective pressures and has driven the evolution of unique adaptive traits in NMRs. Mammalian cells are not usually hypoxia-resistant, requiring uninterrupted $O_2$ availability for survival. Fluctuations in $O_2$ availability can lead to ischaemia/reperfusion injury and irreversible organ damage such as is observed following a heart attack[14]. Given the absence of cardiovascular disease in NMRs[15-18], despite regular fluctuating exposure to hypoxia/anoxia and normoxia, NMR hearts appear to have

evolved resistance to both reduced $O_2$ availability and ischaemia/reperfusion (I/R) injury. NMR metabolism has unusual features, such as the ability to switch from glucose to fructose-driven glycolysis in the brain during anoxia[12]. However, the mechanisms that underpin the extraordinary physiological adaptation to limited $O_2$ availability in the heart are unknown. To determine how these adaptations arise in NMR, we hypothesised that comparison to other African mole-rat genera would enable us to infer the changes in gene expression and metabolic signatures that contribute to the extreme hypoxia tolerance, resistance to cardiovascular injury, and longevity of NMRs.

To do this, we performed RNAseq, metabolomics and pathway enrichment analysis on cardiac tissue from NMR (*Heterocephalus glaber*) and from seven other African mole-rat genera, Cape mole-rat (*Georychus capensis*), Cape dune mole-rat (*Bathyergus suillus*), Common mole-rat (*Cryptomys hottentotus hottentotus*), Natal mole-rat (*C. h. natalenesis*), Mahali mole-rat (*C. h. mahali*), Highveld mole-rat (*C. h. pretoriae*) and Damaraland mole-rats (*Fukomys damarensis*) representing differing burrow and soil types, degrees of sociality, lifespan and hypoxia tolerance.

Morphological and life history characteristics of the African mole-rat genera used in this study are summarised in Supplementary Table 1. In addition, we include the evolutionarily highly divergent Hottentot golden mole (*Amblysomus hottentotus*), an Afrotherian subterranean, solitary mammal. The African mole-rats and the Afrotherian clade last shared common ancestry around 100 million years ago[8] (Fig. 1B). Therefore, the golden mole outgroup comparison should highlight similar adaptations to the subterranean niche. Moreover, the Hottentot golden mole is sympatric with *Cryptomys* and can sometimes utilise

their abandoned burrows, thus offering an outgroup that is subject to similar environmental conditions.

We show that unique cardiac metabolic and genetic features characterise the NMR heart compared to other African mole-rat genera and also to evolutionarily divergent mammals, which result in tolerance to hypoxia/anoxia and also lead to negligible tissue damage by I/R injury. These adaptations include supra-physiological glycogen content providing a readily available biochemical energy source via carbohydrate catabolism during anoxia/hypoxia, enhanced intracellular energy reserves (via increased creatine), and constitutive stabilisation (i.e., non-degradation) of HIF-1α under normoxic conditions. In addition, they show reduced succinate levels post ischaemia, which is a key metabolic driver of reactive oxygen species (ROS), and lower lactate production. NMRs express genetic adaptations at the mitochondrial level, which help to dampen ROS-related damage caused by fluctuating environmental $O_2$ availability.

## Results

### NMRs have evolved a distinct cardiac gene expression

NMRs have evolved a cardiac gene expression profile that is distinct from any of the other mole-rat genera, as well as from the outgroups golden mole (Fig. 2A, B) and the C57/BL6 mouse (Fig. 2D), indicative of extensive adaptations. This is evidenced by separate clustering of NMR transcriptomic data in the hierarchical cluster dendrogram of differential gene expression (Fig. 2A) and in the heatmap summary of predicted differentially expressed genes (DEGs) with the ability to encode transcription factors (Fig. 2B).

Analysis of the 30 most abundantly expressed genes in NMR hearts vs C57/BL6 (Fig. 2C, D; Supplementary Data 1) highlights the importance of adaptations in energy metabolism. In NMR, the majority, including the 10 most abundantly expressed, are mitochondrial oxidative phosphorylation complex genes (*Cox1, Cox3, Cox2, Nd2, Atp6, CytB, Nd1, Nd4, Nd3, Nd5, Nd6*, Fig. 2C). In contrast, in C57/BL6 mouse (*Mus musculus*) hearts the most abundantly expressed genes are responsible for myocardial contractile apparatus (i.e., myosin chains) and oxygen supply (i.e., haemoglobins) (e.g., *Hba a1, Myl2, Hbb-b1, Actc1*, Fig. 2D).

Comparison of the top 30 expressed cardiac genes across all the species analysed exemplifies the unique gene expression pattern of NMRs and their divergent phenotype when compared to other mole-rats, and also to the mouse and golden mole outgroups.

The top 10 expressed genes in the NMR do not appear within the top 30 from any other species, while many of the top 10 genes from the other species are shared in common (Supplementary Data 1).

Pairwise comparisons of the DEG from the RNAseq data among the NMR and the other subterranean genera are shown in the volcano plots in Fig. 3 (see Supplementary Data 3 for a full list of DEGs in NMRs versus other mole-rats and the golden mole). Consistently, higher expression of mitochondrial genes for the electron transport chain enzymes (OXPHOS) is observed in the NMR compared to the other subterranean species we examined. The mRNA levels of the following genes are all elevated: *Mettl12* (mitochondrial matrix methyltransferase protein 12)[19], *Cox1, Cox2,* and *Cox3* (Cytochrome C oxidase, Complex IV); *Nad1-Nad6* (NADH dehydrogenase 2; Complex I); *CytB* (Cytochrome-b; Complex III) and *Atp6* (ATP synthase). Interestingly, Complex II (e.g., *Sdha-d*) was not differentially expressed in NMRs (Fig. 3, Supplementary Fig. 1). Protein expression of complexes I, II, III and V was measured by western blot analysis (Supplementary Fig. 1). Expression levels of complexes I, II and V were not found to be significantly different between NMR and C57/BL6 mouse while expression of complex III was elevated in NMR compared to C57/BL6 mouse.

We have also performed additional NMR versus C57/BL6 mouse DEG analysis (Online Supplementary Fig. 2). Of the 18,012 NMR genes assessed we could analyse 12,206 corresponding mouse genes. Out of these, we were able to calculate log fold changes for 7949 genes that were expressed in both the mouse and the NMR (Supplementary Data 4). Of the top 20 expressed genes in the NMR, only four are also expressed in the mouse, and none of the top 12 in the NMR are also expressed in the mouse. The gene *Clgn* has the highest log fold change in NMRs versus the mouse, but it is ranked 1760 in genes ordered by expression (FPKM) in the NMR.

### NMRs have evolved a distinct cardiometabolic profile

To better understand the putative comparative genetic adaptations, untargeted metabolomic profiling was performed by high-resolution $^1$H magnetic resonance spectroscopy (MRS) on cardiac tissue from the NMR and from the other members of African mole-rat genera, as well as from wild-type C57/BL6 mice (Fig. 4). NMRs showed a distinct cardiac metabolomic profile compared to other African mole-rat genera, and also to the Hottentot golden mole and mouse hearts, as shown in heatmap and hierarchical cluster summary (Fig. 4A). Principal Component Analysis (PCA) and Linear Discriminant Analysis (LDA) of cardiac metabolomic data showed clear separation between three groups (Supplementary Fig. 3, Fig. 4B), with the NMR being distinct, the other African mole-rat genera clustering together suggesting they are metabolically far more similar to each other than they are to NMRs.

The most notable metabolic difference observed between NMR and all other species was a significantly elevated myocardial concentration of glycogen (Fig. 4C), and of glucose-1-phosphate (Fig. 4D), which results from glycogen degradation; these were lower or undetectable (by $^1$H MRS) in the other mole-rat species and undetectable in the mouse heart.

Plasma metabolite analysis shows no evidence of increased insulin or circulating carbohydrates in the NMR (Supplementary Fig. 4).

Thus, increased myocardial glycogen content in NMRs was not driven by altered exogenous metabolite or hormone availability, and instead suggests a much greater capacity for storing myocardial glucose, which can later be metabolised to sustain ATP production. NMRs also had a 44% elevation in myocardial creatine content compared to other mole-rat genera and C57/BL6 mouse ($p < 0.01$, Fig. 4E). This is a remarkable increase in a potential energy reserve pathway in a healthy, non-transgenically modified heart.

### NMRs display constitutive stabilisation of HIF-1α in normoxia

Another interesting finding from our study is the evidence of stabilised (non-degraded) cardiac HIF-1α protein expression in NMRs during normoxia (Fig. 5A–C, representative blots Supplementary Fig. 5). Quantification of HIF-1α protein expression was carried out using three different antibodies against three different protein epitopes (Fig. 5) and was found to be significantly increased in NMR vs C57/BL6 mouse hearts (52% increase, average across three antibodies, Fig. 5A–C, $P < 0.001$). Stabilisation of HIF-1α expression under normoxia is surprising and contrasts with other organisms and tissues under normal $O_2$ availability.

### NMRs show resistance to ischaemia reperfusion injury

In order to examine the physiological significance and functional translation of the genetic and metabolic differences in NMR (Fig. 6A), isolated hearts were subjected to 20 min ischaemia followed by 2 hours of reperfusion (Fig. 6B). At baseline, NMRs maintained lower cardiac function as evidenced from significantly lower left ventricular developed pressure (LVDP $30.2 \pm 2.4$ vs. $73.4 \pm 2.7$ mmHg, $P < 0.0001$, Supplementary Fig. 6) and heart rate (HR $134 \pm 23$ vs. $237 \pm 33$ beats/minute $P < 0.05$, Supplementary Fig. 6) compared to C57/BL6 mouse hearts. This is consistent with in vivo echo and MRI functional assessment and may be an ecophysiological adaptation to life in an energetically taxing environment[15].

After 20 minutes total global normothermic ischaemic anoxia, NMR hearts had enhanced energy supply (29% higher total adenine nucleotide pool, 62% glucose, 52% lactate, Fig. 6E–G) and reduced

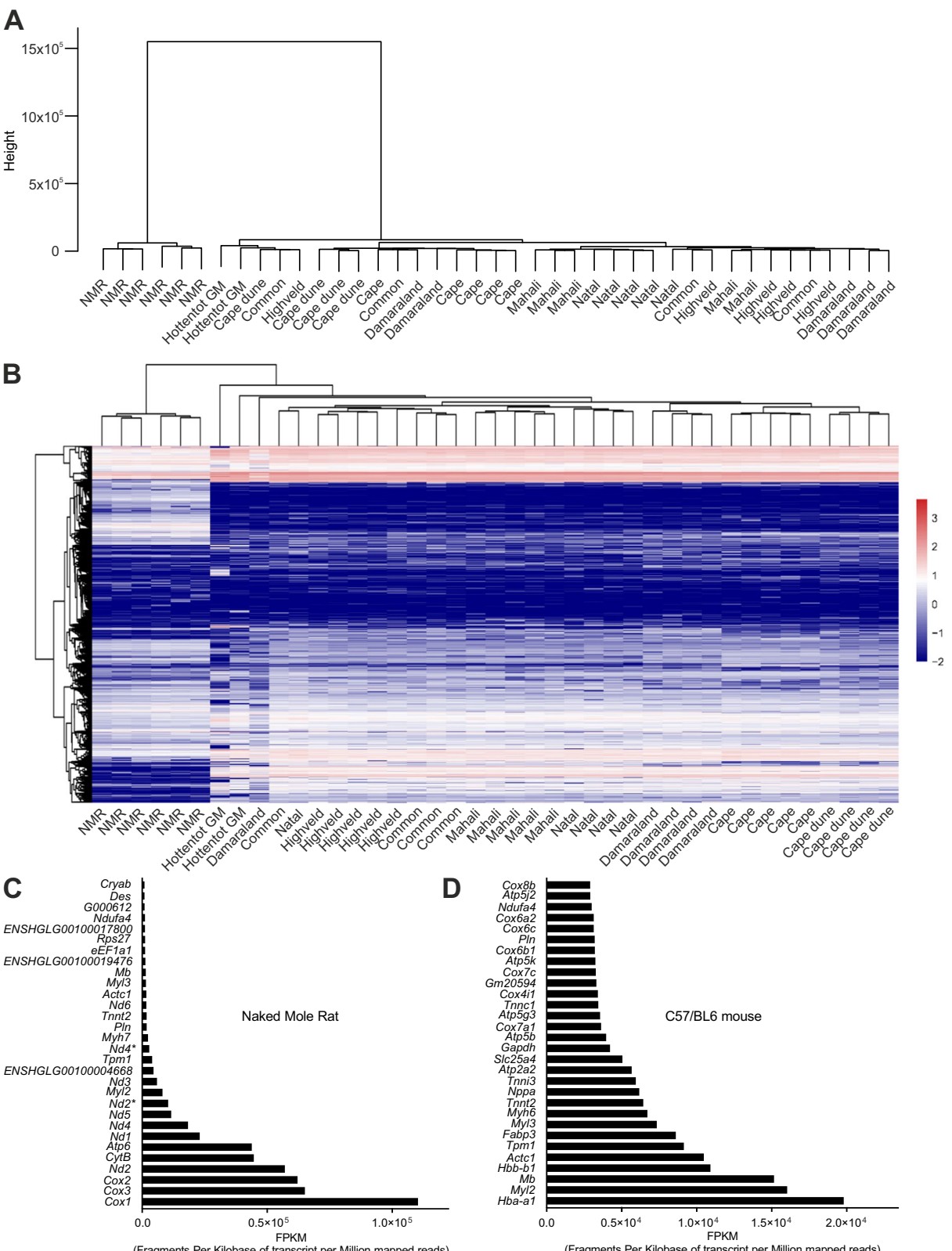

**Fig. 2 | Naked mole-rats exhibit distinct cardiac gene profile expression unrelated to any of the other subterranean genera highlighting genetic adaptations at the mitochondrial level. A** Hierarchical clustering of samples by similarity based on the expression level of all genes. **B** Heatmap with predicted differentially expressed genes encoding transcription factors (TF); *X* axis represents each comparing sample (relative to the NMR), *Y* axis represents DEGs. Colouring indicates the log2 transformed fold change (high: red, low: blue). **C** Normalised comparison of the 30 most expressed genes in NMR hearts and (**D**) C57/BL6 mouse hearts—expressed as fragments per kilobase of transcript per million mapped reads. Nd2* and Nd4* in (**C**) are smaller novel transcripts (BGI_novel_G000346 and BGI_novel_G001276 respectively; Supplementary Data 2). Source data are provided as a Source Data file.

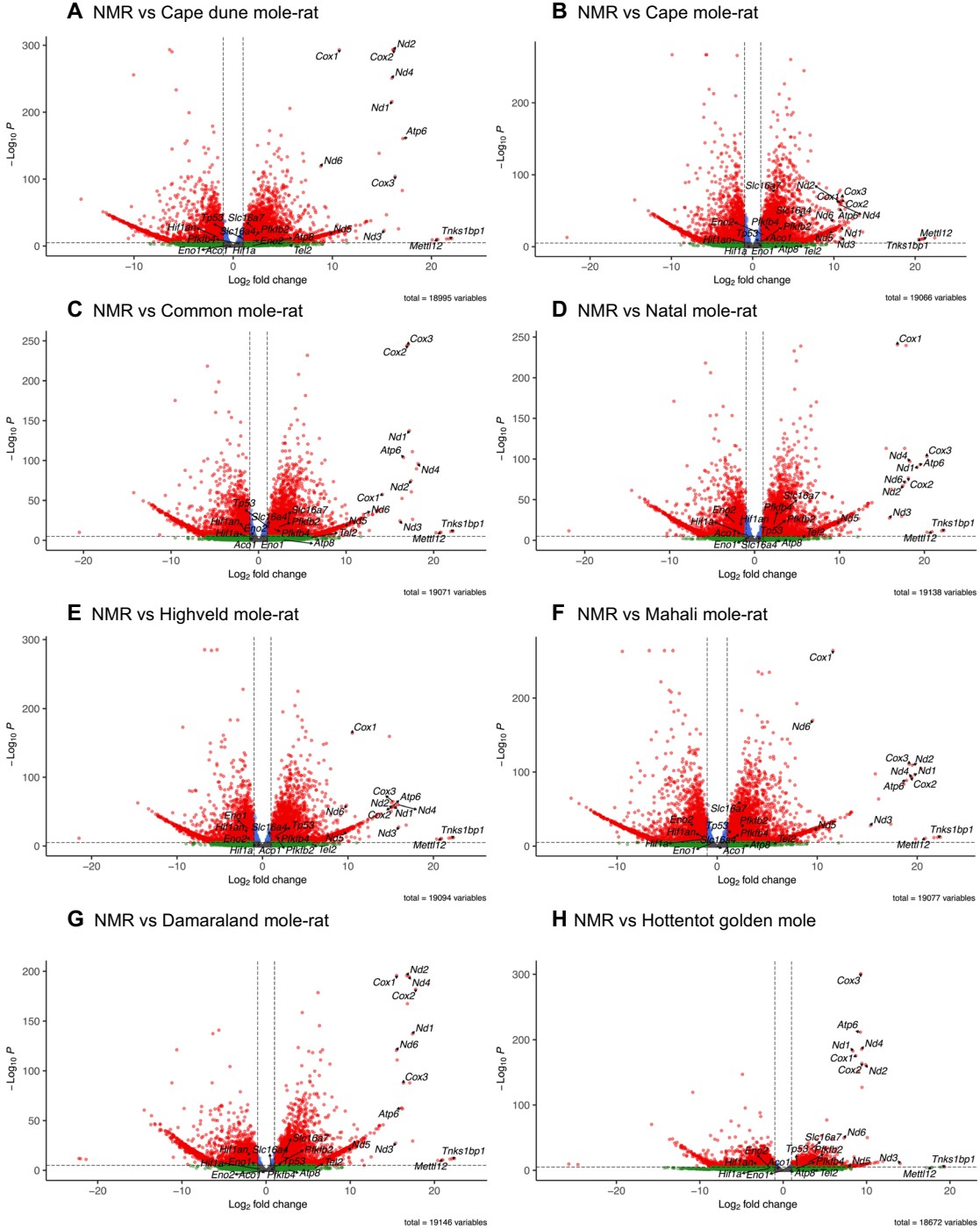

**Fig. 3 | Volcano plot showing differentially expressed genes. A–H** show each respective pairwise comparison between the naked mole-rat (NMR) and the other species in the RNAseq dataset. *Y* axis shows statistical significance as the $-\text{Log}_{10}P$ values while the *x* axis the $\text{Log}_2$ fold change in gene expression (positive values represent overexpression in the NMR). Relevant genes that were identified and discussed in the text are labelled. The default cut-off for $\log_2$FC is >|2| and the default cut-off for *P* value is $10^{-6}$, denoted by the dotted lines. Points are coloured as follows: grey (not significant $P > 10^{-6}$, $\log_2$FC < |2|), green (not significant $P > 10^{-6}$, $\log_2$FC > |2|), blue (significant $P < 10^{-6}$, $\log_2$FC < |2|), red (significant $P < 10^{-6}$, $\log_2$FC > |2|). Source data are provided as a Source Data file.

succinate levels (54% lower, Fig. 6I), the metabolic source of ROS in I/R damage. Succinate efflux 1 min post-ischaemia was also significantly lower (Fig. 6J). NMR hearts showed minimal damage post-20-minute global normothermic ischaemia as infarcted scar tissue was barely detectable unlike in mouse hearts which exhibited extensive tissue damage (Fig. 6C, D). Biochemical analysis has shown that NMR hearts efflux less lactate at baseline during normoxia as well as at 1 min and 1 h post reperfusion of ischaemic tissue, compared to C57/BL6 mice (Fig. 6H). However, during ischaemia NMR hearts have elevated

intracellular lactate compared to mice (Fig. 6G) likely due to increased availability of carbohydrate reserves (glycogen, Fig. 4C) which is broken down to yield increased supply to glycolysis (Fig. 6F) and hence increased endogenous lactate arising from glycolysis (Fig. 6G).

## Discussion

RNAseq and metabolomics were performed on cardiac tissue from NMR and from seven other African mole-rat genera, the evolutionarily divergent Hottentot golden mole and the C57/BL6 mouse. A plethora

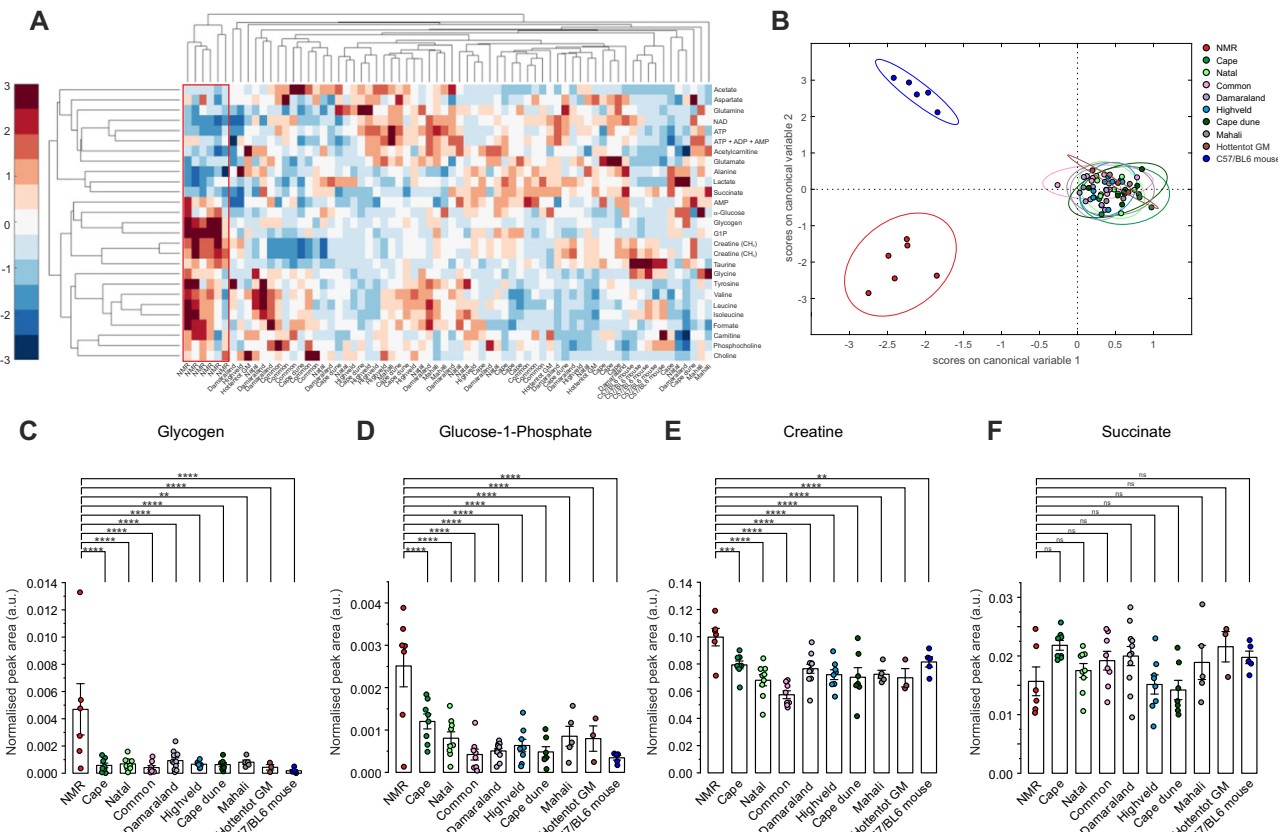

**Fig. 4 | Enhanced cardiac energy reserve is unique to naked mole-rat.**
**A** Heatmap summary of the cardiac metabolite concentrations determined by high-resolution ¹H nuclear magnetic resonance spectroscopy showing distinct profile of naked mole-rat (NMR, n = 6) versus Cape mole-rat (n = 8), Natal mole-rat (n = 9), Common mole-rat (n = 8), Damaraland mole-rat (n = 11), Highveld mole-rat (n = 8), Cape dune mole-rat (n = 7), Mahali mole-rat (n = 5), Hottentot golden mole (n = 3) and C57/BL6 mouse (n = 5), where n correspond to biological repeats. The colorbar denotes the number of standard deviations above or below the row mean of each metabolite. Dendrograms top and left side show hierarchical cluster analysis of related genotypes (columns) and related metabolites (rows). Data are standardised along each row (metabolite) so the mean is 0 and the standard deviation is 1.

**B** Linear discriminant analysis of the metabolite data in **A** showing clear separation of NMR from all other genera. Ellipses correspond to 95% confidence intervals. Unsupervised principal component analysis (PCA) plot is shown in Supplementary Fig. 2. **C** NMR is characterised by supra-normal myocardial glycogen content, **D** elevated glucose-1-phosphate, the product of glycogen catabolism, **E** elevated cellular energy reserve creatine, while **F** succinate was not significantly different under baseline normoxic conditions. P values calculated from a one-way ANOVA of each pairwise comparison with Dunnett's multiple comparisons test. Data are presented as mean ± SEM, *p < 0.05, **p < 0.01, ***p < 0.001, ****p < 0.0001. Source data are provided as a Source Data file.

of metabolic and genetic characteristics were unique to the NMR (Fig. 7). Principal component and hierarchical cluster analysis of RNAseq data uniquely separated NMR data from all other groups. There is a significant upregulation of mitochondrial genes for much of the electron transport chain enzymes including Complex I, III, IV, as well as those for the ATP synthase. Interestingly, succinate dehydrogenase was unaltered across genera. Additionally, *Mct2*, *Pfkfb2*, several glycolytic genes and the master tumour suppressor gene *Tp53* were also consistently relatively upregulated (Fig. 3). Taken together, these genetic adaptations are unique to NMR and not shared with other members of the African mole-rat genera, thus, reflecting adaptive traits within the heart genome unique to the lifestyle of NMRs.

Principal component and discriminant analysis of metabolomic data also uniquely clustered NMR apart from all other groups. Compared to the other genera, metabolic adaptations included supra-normal glycogen content and G-1-P, arising from enhanced glycogen turnover, and a greater total adenine nucleotide pool, thus potentially stabilising available biochemical energy supply via carbohydrate catabolism during stress. The concentration of myocardial glycogen in NMRs is higher than other mammals and was higher than measured in the post-prandial liver where its concentration of this metabolite is the highest (NMR heart glycogen 1.2 ± 0.02 versus post-prandial mouse liver glycogen 0.4 ± 0.08 µmol g⁻¹; p < 0.001)[20]. In addition to ATP

provision, recent evidence suggests that glycogen may have a more dynamic and influential role in coupling myocardial metabolism with contractility, possibly by regulating glycogen-selective autophagy (glycophagy)[21–26]. Whilst reports of enhanced myocardial glycogen content due to impaired glycophagy in diabetic hearts have been linked to impaired diastolic function, there is no evidence of such aberrant effects due to enhanced glycogen content in the NMR heart.

NMR also showed enhanced ability to transiently buffer energy reserves through elevated creatine. It has previously been shown that elevated creatine and glycogen are protective during myocardial ischaemia as they enahance energy reserves[27,28]. Increased myocardial creatine in creatine transporter overexpressing mice has been shown to be beneficial as it protects against I/R injury by decreasing necrosis in a dose-dependent manner[27,28]. Transgenic hearts with elevated creatine exhibited improved functional recovery following ex vivo I/R (59% of baseline vs. 29% of baseline cardiac function)[27,28]. Furthermore, cellular creatine loading has been shown to delay mitochondrial permeability transition pore opening in response to oxidative stress, suggesting an additional mechanism to prevent I/R injury[27,28]. However, in other mammals it is not possible to increase myocardial creatine concentrations to supra-normal levels because it is subject to tight regulation by the sarcolemmal creatine transporter[27,28]. Furthermore, artificial increases in myocardial creatine content by

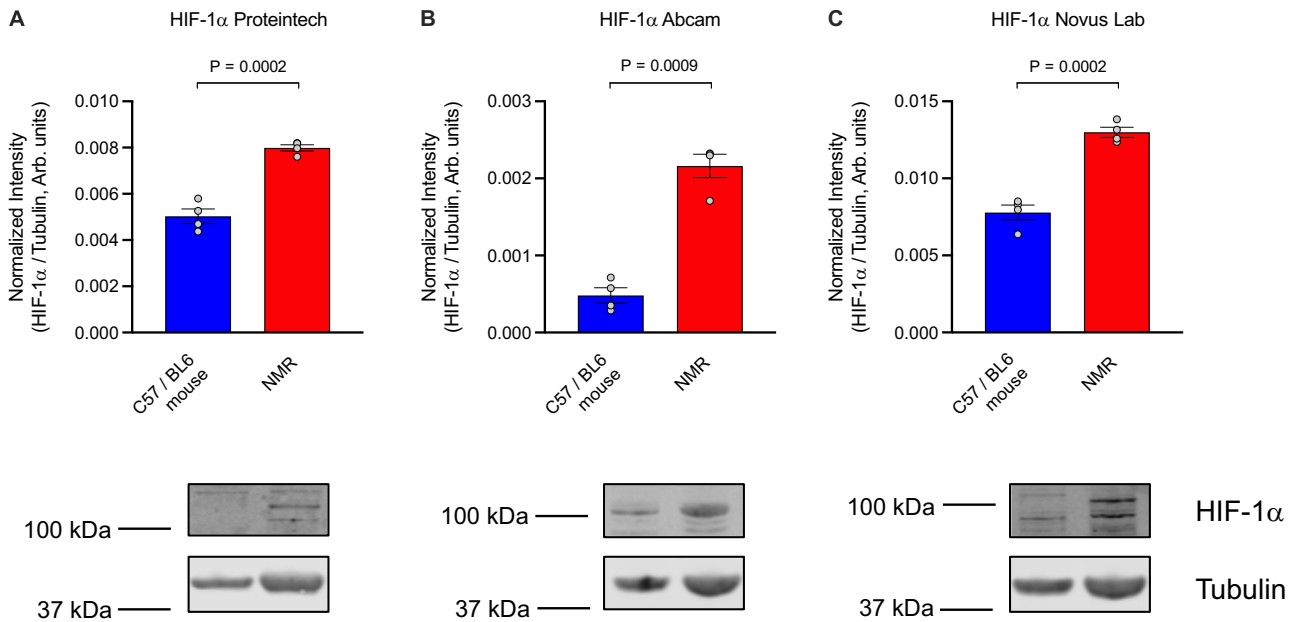

**Fig. 5 | NMRs display constitutive stabilisation of HIF-1α under normoxic conditions.** Quantification of HIF-1α protein expression using different antibodies against three different protein epitopes (Representative western blots in Supplementary Fig. 5). **A** Proteintech rabbit polyclonal anti HIF-1α antibody (Clone developed against protein sequence including amino acids 574–799 of the human HIF-1 alpha protein; GenBank: BC012527.2). **B** Abcam rabbit polyclonal anti-HIF-1α antibody (Clone developed against recombinant fragment. This information is proprietary to Abcam and/or its suppliers). **C** Novus Lab rabbit polyclonal anti-HIF-1α antibody (Clone developed against a fusion protein including amino acids 530–825 of the mouse HIF-1 alpha protein; Uniprot #Q61221). Experimental groups contain $n = 4$ animals and western blot analysis was performed using several technical replicates. HIF-1α protein expression normalised to tubulin. Statistical analysis was performed using unpaired *t* test (with Welch's correction and two-tailed *P* values), and data are represented as mean ± SEM. Source data are provided as a Source Data file.

transgenesis have also shown that the hearts of other mammals are incapable of maintaining the augmented creatine pool adequately phosphorylated, resulting in increased free ADP levels, LV hypertrophy, and significant dysfunction[28]. Taken together, metabolic adaptations are unique to NMR and not shared with the other members of African mole-rat genera, thus, these adaptations also do not construe mammal-wide convergence in adaptive traits within the heart metabolome.

The PCA analysis of our metabolomics data indicates a high degree of similarity between the other genera of mole-rats (and the golden mole) compared to NMR, regardless of whether they are eusocial, solitary or social. This suggests that the evolutionary pressures imposed on these other subterranean species are distinct from those experienced by the NMR. Despite group living and multiple occupancy of nest chambers in other species of mole-rat, it seems that the NMR faces more extreme challenges due to absolute social group size, burrow size, and possibly the distance from the nearest mole-hill that determines gas exchange.

NMR hearts showed a significant elevation of HIF-1α levels under normoxic conditions compared to the C57/BL6 mouse. This was confirmed at the protein level using several different antibodies raised against different HIF-1α epitopes to best compare two different animal species. This phenomenon contrasts with what is currently known about hypoxia/anoxia-mediated degradation of HIF-1α protein. HIF-1α is an oxygen-dependent transcriptional activator that mediates a wide range of adaptations to reduced oxygen. Under normoxia HIF-1α expression in NMR is constituent[29,30] and stable due to the mutation in HIF-1α (T407 to I) and in Von Hippel–Lindau (VHL; V166 to I) that both synergistically prevent the ubiquitin-mediated degradation of HIF-1α[29,30].

Unlike humans, who are prone to heart injury by hypoxia and anoxia caused by blood occlusion during heart attacks, NMR hearts have adapted to evade such damage. Ischaemia or hypoxia typically leads to elevated levels of succinate in the heart, and other tissues, and is a key driver for the production of ROS during reperfusion[14]. In the hearts of NMR subjected to ischaemia, we observed lower levels of succinate than in the mouse hearts. Low succinate levels may limit ROS production during reperfusion and therefore mitigate tissue damage. Increased myocardial succinate levels during ischaemia arise at least in part from the reversal of succinate dehydrogenase, driven by fumarate/malate generated during ischaemia and partial reversal of the malate/aspartate shuttle. Upon reperfusion, the accumulated succinate is rapidly re-oxidised by succinate dehydrogenase, driving extensive ROS generation by reverse electron transport (RET) at mitochondrial complex I[14].

However, increased succinate level is also an inhibitor of prolyl hydroxylase (PHD), and therefore reduced succinate levels may also decrease inhibition of PHD, leading to weak stabilisation of HIF-1α and thus its enhanced degradation during ischaemia. Interestingly, a blunted HIF-1α response in anoxia is also observed in human tissues that experience regular variation of $O_2$ such as anoxic cord blood monocytes[31]. In keeping with the reduced succinate levels, NMR hearts showed negligible tissue damage post-ischaemia reperfusion, while in contrast, the C57/BL6 mouse showed extensive tissue damage due to reperfusion injury leading to elevated myocyte necrosis.

At present, it is unknown whether a species such as the mouse, which is not protected from ischaemia, effluxes lactate at the same rate as NMR. Our finding of lower lactate efflux in NMR heart may suggest improved use of lactate as fuel and thus could potentially contribute to the resistance to I/R injury in this species. How naked mole-rats deal with lactate accumulation during I/R, thus avoiding lactic acidosis, is, therefore, a fascinating phenomenon that future work may determine contributes to cardiac protection in NMRs under ischaemia.

It has also been shown that the mitochondria from a whole host of tissues (skeletal muscles, diaphragm, heart, spleen, brain, lung kidney) from mice, bats, and NMRs contain two mechanistically similar

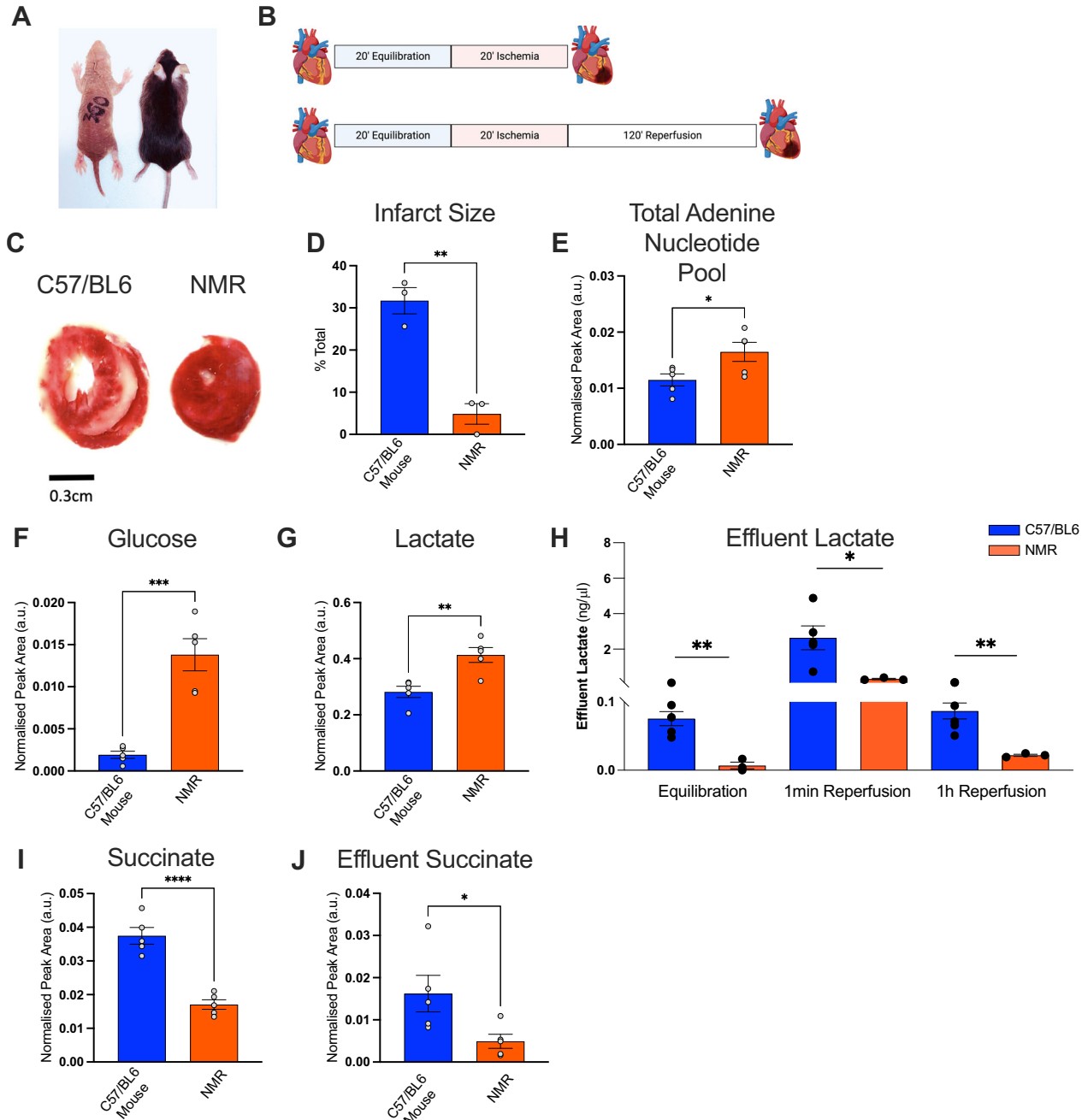

**Fig. 6 | Naked mole-rat heart is resistant to ischaemic damage. A** Representative naked mole-rat and C57/BL6 mouse used in the study. **B** Langendorff heart ischaemia/reperfusion protocols, figure created with Biorender.com. **C** Images of representative cardiac sections from C57/BL6 mouse and NMR post I/R. Red colour-viable tissue, white areas- post-infarct scar. **D** Reduced infarct size in NMR vs C57BL6 post-myocardial ischaemia (20 min) and reperfusion (120 min) protocol **P = 0.0025. [1]H nuclear magnetic resonance spectroscopy analysis of cardiac metabolites post-20-minute ischaemia in NMR vs C57/BL6 mouse, **E** elevated total adenine nucleotide pool (sum of ATP + ADP + AMP) *P = 0.039 two-tailed *t* test, **F** elevated myocardial glucose ***P = 0.0003 two-tailed, **G** elevated myocardial lactate **P = 0.0043 (**H**) reduced effluent lactate **P = 0.003 *P = 0.0439 two-tailed (**I**) reduced myocardial succinate ****P = 0.0001 and **J** reduced effluent succinate (1 min reperfusion) *P = 0.041. *n* = 5/group, Data mean ± SEM. *t* test two-tailed, data normality tested by Shapiro–Wilk. Source data are provided as a Source Data file.

systems to prevent ROS generation: mitochondrial membrane-bound hexokinases (I and II) and creatine kinase. Specifically, both of these metabolic systems function in a way that one of the kinase substrates (i.e. mitochondrial ATP) is electrophoretically transported by the ATP/ADP antiporter to the catalytic site of either bound hexokinase or creatine kinase without affecting ATP concentration in the cytosol. ADP, another kinase reaction product, is transported back to the mitochondrial matrix via the antiporter by the same electrophoretic process. This system continuously supports ATP synthase independent of glucose and creatine availability. This unique ATP

homoeostatic mechanism would be supported by the elevated creatine and glycogen observed in NMR hearts at baseline (normoxic conditions) making them available for ATP provision during times of stress (ie. ischaemia)[32]. These conditions keep mitochondria in a state of mild depolarisation with the membrane potential Δψ maintained at a lower than maximal level[32], sufficient to completely inhibit mROS generation. During ageing in C57/BL6 mice (2.5 years), mild depolarisation disappears in most tissues, including the heart, however, age-dependent decreases in the levels of bound kinases are not observed in NMRs[32]. As a result, protein damage, which is substantial during the

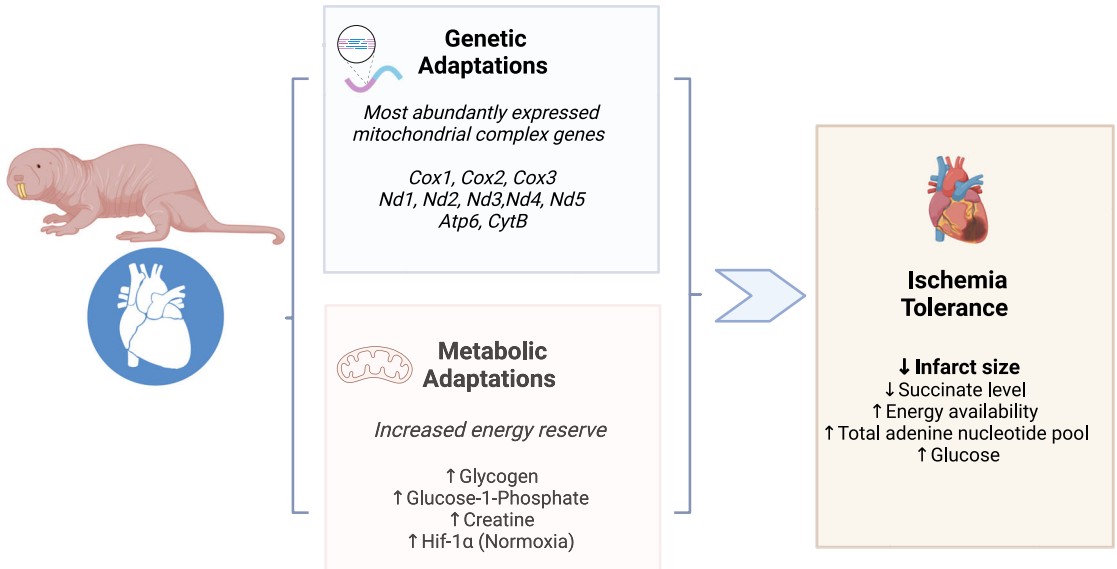

**Fig. 7 | Naked mole rats have distinctive cardiometabolic and genetic adaptations to their underground low-oxygen lifestyles-schematic summary.** Figure created with Biorender.com.

ageing of short-lived mice, is stabilised at low levels during the ageing of long-lived NMRs. It has been suggested that mild mitochondrial depolarisation is a crucial component of the anti-aging system[32] contributing to NMR longevity. We also observed that highly expressed mitochondrial genes in NMRs do not necessarily translate to enhanced mitochondrial protein expression and respiration. The relatively low basal cardiac function and mitochondrial respiration could be part of ecophysiological adaptations to life in an energetically taxing environment and may also contribute to NMR longevity.

In the current study, we have investigated differences between C57/BL6 mice, the golden mole, and mole-rats, including NMRs under uniform basal $O_2$ conditions (normoxia) and carried out cardiac perfusions at 37 °C for C57/BL6 mice and NMRs. Thus, a potential limitation of our study is that it was performed in conditions not representative of the NMR environment. Nevertheless, our study provides insights into the evolution of the NMR's remarkable cardiac adaptations to life in hostile conditions at low $O_2$. Collectively, they may contribute to their exceptional longevity and resistance to ischaemic pathologies. The extreme metabolic and genetic traits we identified offer opportunities for advancing other areas of physiological and medical research, including the development of novel therapeutic approaches.

## Methods

Our research complies with all the relevant ethical regulations. The Animal Use and Care Committee of the University of Pretoria evaluated and approved the experimental protocol and collection of all samples (ethics clearance number: NAS022/2021), with DAFF section 20 approval (SDAH-Epi-21051907211). Animal experiments at QMUL were approved by the local ethical review committee and in accordance with the Home Office Animals in Scientific Procedures Act 1986.

### Animals

NMRs were bred at the School of Biological and Behavioural Sciences, Queen Mary University of London. The non-breeding adult NMRs used in this study were second-generation or more captive-born, descended from animals captured in Kenya in the 1980s. Colonies were maintained using artificial burrow systems (group-housed in interconnected multi-cage systems) at 30 °C and 21% $O_2$ in 50% humidity with a 12 h light cycle. Their diet consisted of fresh vegetables, fruit,

and tubers (sweet potatoes) *ad libitum*. All water requirements were obtained from the food resources. A noticeable drop in the heart-to-body mass ratio is observed in African mole-rats after a year in captivity. At least one year of captivity has resulted in the wild-caught and captive-bred African mole-rat species having a similar heart-to-body mass ratio. Both sexes (mole-rats) were used in the study, as summarised in Supplementary Table 1). The ages selected for this study allowed for physiological age matching such that all animals were at equivalent percentages of maximum lifespan and, therefore, not the same chronological age. C57/BL6 (J strain, male) mice were purchased from Charles River Laboratories, UK and kept in individually ventilated cages, with access to food (PicoLab Mouse Diet 20 EXT 5R58, irradiated, I-DIET-5R58-9KG-BG, IPS) and water ad libitum. All mice are housed in IVCs (Individual Ventilated Cages). Enrichment is provided in the form of tunnels and chew sticks. Cages are cleaned out under a LEV hood when required, the maximum time between cage changes is 14 days.

*Georychus capensis*, *Bathyergus suillus*, *Cryptomys hottentotus hottentotus*, *C. h. pretoriae*, *C. h. mahali*, and *C. h. natalenesis* were wild-captured in South Africa using Hickman live traps, baited with a small piece of sweet potato. All traps were monitored for captures every 2–3 hours over the course of the day and left overnight, being checked first thing in the morning. Permission to capture these species was obtained from all landowners, and a collecting permit was obtained from the relevant nature conservation authorities (Permit number: Western Cape- CN44-87-13780, Gauteng- CPF6-0124, Kwa-Zulu Natal-OP1545/2021). *Fukomys damarensis* were laboratory maintained. Captured animals were brought back and individually housed at the University of Pretoria at ~27 °C and 21% $O_2$ in 50% humidity with a 12 L:12D light cycle. All animals were housed in large polyurethane crates (1 × 0.5 × 0.5 m) with wood shavings and paper towelling for nesting material in temperature-controlled rooms. All wild-caught African mole-rat species were maintained in captivity for at least one year before tissue harvesting. Due to difficulty in maintaining the Golden moles in captivity and their short lifespan, they were kept captive for ~3 weeks prior to tissue harvest. Each animal was terminally anaesthetised using isoflurane. Immediately upon cessation of breathing, heart was dissected and within 60 seconds placed in cryovial containing RNA later, and flash-frozen in liquid nitrogen. Subsequently, samples were placed in the −80 °C freezer for storage.

## RNAseq/transcriptome analysis

Total RNA extraction and analysis were performed commercially by BGI Tech solutions (Hong Kong) using sequencing platform DNBseq and read length PE 100 bp. We sequenced 41 samples as follows: Naked mole-rat, *Heterocephalus glaber* ($n = 6$), Cape mole-rat, *Georychus capensis* ($n = 5$), Cape dune mole-rat, *Bathyergus suillus* ($n = 4$), Common mole-rat, *Cryptomys hottentotus hottentotus, C. h. natalenesis* ($n = 5$), Mahali mole-rat, *C. h. mahali* ($n = 5$), Highveld mole-rat *C. h. pretoriae* ($n = 5$), Damaraland mole-rat, *Fukomys damarensis*, Hottentot golden mole, *Amblysomus hottentotus* ($n = 2$) and C57BL6 mice *Mus musculus* ($n = 6$). An average of 4.28 Gb of sequence was generated per sample.

Standard bioinformatic processing was performed as a service by BGI Tech Solutions. Briefly, the reads mapped to rRNAs were removed to produce rawdata, then low-quality reads were filtered where more than 20% of the qualities of the base were lower than 10, reads with adaptors and reads with unknown bases (*N* bases >5%) were also removed to produce "clean reads". Clean reads were mapped onto reference genome (*Heterocephalus glaber* Ensemble_release-108). Genome mapping was conducted using HISAT2 (Hierarchical Indexing for Spliced Alignment of Transcripts; http://www.ccb.jhu.edu/software/hisat).

The average mapping ratio with the reference genome was 18.13%, while the average mapping ratio for genes was 32.28%; 20,285 genes were identified. Mapping was followed by novel gene prediction, SNP and INDEL calling, and gene splicing detection. StringTie (http://ccb.jhu.edu/software/stringtie) was used to reconstruct transcripts, while Cuffcompare and Cufflinks tools (http://cole-trapnell-lab.github.io/cufflinks) were used to compare reconstructed transcripts to the reference annotation. Novel transcripts were defined as (i) unknown, intergenic transcripts, (ii) transfrags falling entirely within a reference intron, (iii) generic exonic overlaps with a reference transcript, and (iv) potentially novel isoforms (fragments) where at least one splice junction is shared with a reference transcript. CPC was then used to predict the coding potential of novel transcripts. Novel coding transcripts were then merged with reference transcripts to get a complete reference, and downstream analysis was based on this complete reference. GATK[33] was deployed to call SNPs and INDELs for each sample.

Finally, DEGs were identified between samples with DEseq2[34], and clustering analysis and functional annotations performed. Hierarchical clustering for DEGs was performed using pheatmap, a function of R. Following Gene Ontology Analysis of DEGs GO functional enrichment analysis was undertaken using phyper, a function of R. The analysis pipeline is summarised in Supplementary Fig. 8.

Due to the limited mapping rate using the naked mole-rat as a reference for the comparison with the mouse C57/BL6 transcriptome (and vice versa), and the limited common expression of the most expressed genes in each species, it was not possible to do the same range of analyses for the mouse versus the naked mole-rat (or mouse versus the other species), as for the naked mole-rat versus the subterranean species listed above. Instead, as a workaround, we identified orthologous genes in the naked mole-rat and mouse in order to compare gene expression between the species. For transcripts that mapped to the naked mole-rat genome, we extracted the corresponding gene sequences directly from the naked mole-rat genome (GCA_944319725.1).

For novel transcripts, we used associated protein IDs identified by BGI and downloaded the corresponding gene sequences using the Entrez Direct Utilities[35]. We used the Basic Local Alignment Search Tool (BLAST; v2.11.0[36]) with parameters -evalue 1e-50 -perc_identity 80 to query all sequences against the mouse reference genome (GCF_000001635.27). We took the best hit for each transcript and calculated the log2 fold change between the mean abundance (FPKM) of each species. We plotted the data using the R package ggplot2[35,36].

## Langendorff heart perfusion

Beating hearts were excised from terminally anaesthetised naked mole-rats ($n = 5$) and C57/BL6 mice ($n = 5$) (Charles River, UK) for Langendorff perfusions as previously described[37]. Ex vivo cardiac function data was recorded using ADINstruments Chart software (v6). All perfusions were carried out at 37 °C. After 20 minutes of equilibration, hearts were subject to 20 minutes of global normothermic ischaemia (37 °C)[38]. Hearts were snap-frozen at the end of the protocol using a Wollenberger clamp pre-cooled in liquid $N_2$. Ischaemia/reperfusion (I/R) tissue damage in NMRs was compared to C57BL6 mouse heart. Coronary effluent was analysed using the lactate quantification kit (Sigma Aldrich Merck, UK).

The comparison between the mouse and the NMR was made as they are similar-sized small rodents with widely different tolerance of hypoxia. Furthermore, mouse is widely used due to its anatomical, physiological, and genetic similarity to humans[39]. We were unable to repeat ex vivo cardiac perfusion protocols in all other 5 mole-rat genera hearts as they are not kept in captive colonies in the UK. Blood samples were collected at the time of experimental endpoint into heparinized tubes, centrifuged, plasma flash frozen in liquid $N_2$ and stored in −80 °C until analysis.

Plasma biochemical profiling was carried out by the MRC Mouse Biochemistry Laboratory (Addenbrookes NHS Hospital, Cambridge). Plasma triglycerides (4 μl sample) were measured using Siemens Dimension EXL clinical sample analyser with Siemens Healthcare reagents (DF69A). Enzymatic assay was based on lipoprotein lipase conversion of triglycerides into free glycerol and fatty acids[40]. Plasma glucose was quantified using the automated enzyme assay on the Siemens Dimension EXL analyser (3 μl of sample, reagent manufacturer Siemens Healthcare, DF30). Assay is an adaptation of the hexokinase-glucose-6-phosphate dehydrogenase method[41].

Plasma lactate (8 μl of sample) was analysed using the automated assay (Siemens Dimension EXL analyser, Siemens Healthcare product code DF54). Plasma insulin was analysed using guinea pig Insulin (INS) ELISA (abx150418, Abbexa, USA).

## Myocardial infarct size quantification

Myocardial infarct size was quantified as described previously[38,42]. In brief, after 20 min of equilibration, Langendorff perfused hearts ($n = 3$/ group) were subject to 20 min of global normothermic ischaemia and 2 hours of reperfusion. At the end of the protocol, hearts were perfused for 10 mins with 3% triphenyltetrazolium chloride (TTC) in KH Buffer followed by 10 min incubation in 3% TTC-KH. Tissue was sectioned (mouse heart gauge, Zivic instruments, USA), and infarct field was scanned using HPSmart Colour Laser Scanner Software (V14.1.0) quantified using ImageJ Software (v 1.53t)[38].

## ¹H NMR spectroscopy

Powdered heart samples were subject to methanol/water/chloroform phase extraction[37]. Frozen heart tissue was homogenised in 2 mL each of ice-cold methanol, chloroform, and Millipore water and vortexed. Samples were centrifuged for 1 hour at $2500 \times g$ at 4 °C to separate aqueous, protein, and lipid layers. The upper aqueous phase was separated, 20–30 mg chelex-100 was added to chelate paramagnetic ions, vortexed, and centrifuged at $2500 \times g$ for 1 minute at 4 °C. The supernatant was transferred to a fresh falcon tube containing 10 μL of universal pH indicator solution followed by vortexing and lyophilization. Dual-phase-extracted metabolites were reconstituted in 600 μL of deuterium oxide [containing 8 g/L NaCl, 0.2 g/L KCl, 1.15 g/L Na2HPO4, 0.2 g/L KH2PO4 and 0.0075% w/v trimethylsilyl propanoic acid (TSP)] and adjusted to pH ≈ 6.5 by titrating with 100 mM hydrochloric acid. Coronary effluents were frozen immediately upon collection in N2 and stored at −80 C until the NMR analysis. 100 μl of D2O containing 0.0075% w/v trimethylsilyl propanoic acid (TSP) as internal reference was added to 500 μl of effluent.

$^1$H nuclear magnetic resonance spectra were acquired using a vertical-bore, ultra-shielded Bruker 14.1. tesla (600 MHz) spectrometer with a bbo probe at 298 K using the Bruker noesygppr1d pulse sequence. Acquisition parameters were 128 scans, 4 dummy scans and 20.8 ppm sweep width, acquisition time of 2.6 s, pre-scan delay of 4 s, 90° flip angle, and experiment duration of 14.4 minutes per sample. TopSpin (version 4.0.5) software was used for data acquisition and for metabolite quantification. FIDs were multiplied by a line broadening factor of 0.3 Hz, and Fourier-transformed, phase and automatic baseline-correction were applied. Chemical shifts were normalised by setting the TSP signal to 0 ppm. Metabolite peaks of interest were initially integrated automatically using a pre-written integration region text file and then manually adjusted where required. Assignment of metabolites to their respective peaks was carried out based on previously obtained in-house data, confirmed by chemical shift and using Chenomx NMR Profiler Version 8.1 (Chenomx, Canada). Peak areas were normalised to the total metabolite peak area. Quantification of glycogen was performed by $^1$H magnetic resonance, giving a measure of the concentration of glucose monomers that are present in the observed peak. Being a large macromolecule, with possible differences in the mobility of glycosyl units, glycogen has been reported to be fully visible by MRS. The modified dual-phase Folch extraction method used for separating aqueous and lipid metabolites was not optimised for the extraction of glycogen, however, all samples underwent the same extraction procedure allowing between species comparison.

## Western blotting

Powdered heart samples were homogenised (100 μL of buffer per 10 mg of cardiac tissue) on ice in 100 mM Tris buffer (pH 7.4) supplemented with complete mini EDTA-free protease inhibitor (Roche) using a glass tissue grinder. The tissue homogenates were re-suspended in an equal volume of 2× reducing SDS sample buffer. Proteins were resolved by SDS-PAGE (4–20% Mini-PROTEAN TGX, Bio-Rad or Novex 4–20% Tris-Glycine, ThermoFisher Scientific) using Mini-Protean 3 system (Bio-Rad) or XCell4 SureLock Midi Cell and transferred using semi-dry system (Bio-Rad) to 22 μm PVDF or Nitrocellulose membranes (Bio-Rad). Following the transfer, membranes were blocked for one hour with 5 % non-fat dried milk in TPBS at RT.

After blocking, membranes were immunoprobed with primary antibodies dissolved in 5 % non-fat dried milk in TPBS overnight. Subsequently, membranes were washed three times with TPBS and incubated with IR-dye conjugated secondary antibodies in Intercept blocking buffer (LI-COR) or with horseradish peroxidase-coupled anti-rabbit IgG antibody in 5 % non-fat dried milk in TPBS for one hour at RT in the dark. After the incubation, membranes were washed three times with TPBS and imaged using Odyssey DLx infra-red imaging system (LI-COR) or ECL reagent (ThermoFisher Scientific) and imaging cabinet. Protein bands were analysed and quantified using Image Studio Lite (LI-COR) or ImageJ (NIH).

**List of vendors and dilution of antibodies used in this study.** Anti HIF-1α Proteintech (#20960-1-AP); (Clone developed against protein sequence including amino acids 574–799 of the human HIF-1α protein; GenBank: BC012527.2) 1:1000.

Anti HIF-1α Abcam (#ab 179483); (Clone developed against recombinant fragment. This information is proprietary to Abcam and/or its suppliers) 1:1000.

Anti HIF-1α Novus lab (#NB100-479); (Clone developed against a fusion protein including amino acids 530–825 of the mouse HIF-1α protein; Uniprot #Q61221) 1:1000.

Anti β-Tubulin ThermoFisher Scientific (#62204) 1:5000.

Anti α-Actin Merck Millipore (#MAB1501) 1:3000.

Total OXPHOS Rodent WB Antibody Cocktail (#ab110413) 1:1000.

## Statistics and reproducibility

Data are presented as mean ± SEM. Comparison between groups was performed by Student's $t$ test (Gaussian data distribution), two-way analysis of variance (ANOVA) with Bonferroni's correction for multiple comparison, and one-way ANOVA using Bonferroni's correction for multiple comparisons where applicable. Normality of data distribution was examined using Shapiro–Wilk's normality test. Statistical analysis was performed using GraphPad Prism (v9) software. Unclassified PCA was performed using a PCA toolbox in Matlab. Data was auto-scaled prior to PCA calculation using venetian blinds cross-validation and 5 cv groups[43]. All quantified analytes were included in the analysis. Classified discriminant analysis was performed using a classification toolbox in Matlab using linear discriminant analysis, bootstrap validation, and 100 iterations. Hierarchical cluster analysis was performed in Matlab using the function clustergram. No statistical method was used to predetermine sample size, and sample sizes were governed by the availability of rare biological material used in the study. No data were excluded from the analyses; the experiments were randomised; the investigators were blinded to allocation during outcome assessment. Differences were considered significant when $P < 0.05$. DEGs were visualised with enhanced volcano plots implemented in R-Studio[44].

## Reporting summary

Further information on research design is available in the Nature Portfolio Reporting Summary linked to this article.

# Data availability

The RNAseq data generated in this study have been deposited in the ArrayExpress database under accession code E-MTAB-13808. All data generated in this study have been deposited (open access) in the Dryad database, and can be accessed here: https://doi.org/10.5061/dryad.w9ghx3fts and https://doi.org/10.5061/dryad.66t1g1k66. Source data are provided in this paper.

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

## Acknowledgements

We would like to thank Dr. Harold Toms and Dr. Nasima Kanwal QMUL NMR facility for the technical assistance with the NMR spectroscopy and Prof Yannick Wurm for constructive criticism of the study. DA is the recipient of the Wellcome Trust Career Re-entry fellowship (221604/Z/20/Z). This work was supported by: the School of Biological and Behavioural Sciences (HEFCE Lectureship to DA), Bart's Charity (G-002145 DA), BHF MRes studentship (FS/4YPhD/P/20/34016 DA, NP), the Medical Research Council (MC_UU_00014/5 MPM), the Medical Research Council UK (MC_UU_00028/4 MPM) and by a Wellcome Trust Investigator award (220257/Z/20/Z MPM). NCB acknowledges SARChI Chair of Mammal Behavioural Ecology and Physiology (NGUN 64756). J.L.J.M acknowledges Biotechnology and Biological Sciences Research Council BLAST Pump Priming Award G123327. TRE acknowledges support from NIHR Biomedical Research Centre at Guy's and St Thomas' NHS Foundation Trust and KCL; the Centre of Excellence in Medical Engineering funded by the Wellcome Trust and EPSRC (WT 203148/Z/16/Z) and the BHF Centre of Research Excellence (RE/18/2/34213). MRC Mouse Biochem Lab, Cambridge, acknowledges the Medical Research Council (MC_UU_00014/5). Figures 1B, 6B and 7 created with Biorender.com.

## Author contributions

Conceptualisation: D.A., C.G.F., methodology: D.A., C.G.F., T.R.E., investigation: D.A., C.G.F., T.R.E., J.L.M., J.D.G., R.C., N.P., H.A.P., D.H.,

N.B., visualisation: D.A., C.G.F., T.R.E., J.L.M., J.D.G. Funding acquisition: D.A., project administration: D.A., supervision: D.A., writing—original draught: D.A., C.G.F., T.R.E., J.L.M., writing—review & editing: D.A., C.G.F., T.R.E., J.L.M., M.P.M.

## Competing interests

The authors declare no competing interests.
