## [Peer Review File · Nature Communications]

Naked mole rats have distinctive cardiometabolic and genetic adaptations to their underground low-oxygen lifestylesREVIEWER COMMENTS

Reviewer #1 (Remarks to the Author):

This well written manuscript addresses differences in bulk RNAseq of hearts from 9 species of subterranean mammals comparing each of these data sets with that of the naked mole-rat [NMR] and highlights key differences that may contribute to the extraordinary tolerance of NMRs to hypoxic conditions and ischemia /reperfusion [I/R] injury. The authors then follow up with an ex vivo study examining the responses of mice and mole-rats to I/R that further confirms NMRs are indeed more resistant than mice to I/R. These are original novel data that would be gladly received by all working on the comparative biology of subterranean animals as well as those interested in possibly translating these findings into biomedical application relevant to cardiac diseases.

With so large a data set it is understandable that reported findings have been specifically chosen, however their well- developed introduction suggests that something more than what was discussed will be included and further justifies this: For example, surprisingly looking at the volcano plots (Fig 3), the NMR shows the greatest similarity to that of the golden mole! Yet this is not addressed. Equally surprising is the fact that PCA analysis suggests greater similarity amongst the southern African Bathyergids regardless of whether or not they are eusocial, solitary and/or social and therefore subject to different degrees of hypoxia stress at least in the nest area where multiple animals versus a single animal may be resting. Despite including information regarding these features in Fig 1, the impact of their divergent lifestyles on hypoxia tolerance is not discussed. Nor do the authors discuss the results of how gene profiles of their outgroup and that of mice differs from the various mole-rat species, despite the fact that they highlight the importance of the need for an outgroup in the introduction. It would have been nice to see their data on mice gene profiles included in Figure 2B. If RNAseq was not done on mouse hearts in the absence of I/R, there are numerous other studies where this has been done that maybe could be included to show just how different mice and NMR heart gene profiles are.

Could the unique transcriptomic and metabolomic profile of the naked mole-rat infer that the NMR is taxonomically distinct as suggested by Patterson 2014. <https://doi.org/10.1111/zoj.12201> ? Are there potentially other explanations as to why this species is such an outlier.

The second section of the paper uses ex vivo preparations of naked mole-rat and mouse hearts and subjects these to ischemia /reperfusion using standard techniques. The following sentence " Ischemia/reperfusion outcome in NMRs was compared to C57Bl6 mouse heart due to their anatomical, physiological, and genetic similarity to humans", in their methods on Langendorff preparations seems spurious. The data suggest the NMR is very different to C57BL/6 mice in terms of heart size, function and gene profile so are you simply saying you use the mouse as a proxy for humans? Given that both NMRs and humans have far lower heart rates and more similar ejection fractions, this justification seems wrong and is not needed for it is perfectly acceptable to compare the mouse and the NMR as similar sized small rodents with widely different tolerance of hypoxia. The I/R results obtained are terrific and strongly support the findings from the gene profile data.

Figure 6C: I would have liked to see representative cardiac section images from control (normoxia) mice and mole-rats rather than only images post I/R, showing an "infarction".

This MI is not labelled on the figure, so to the uninitiated they just look like lighter images in mice. Having a control group set of images would also enable a better understanding of the changes with I/R treatment in each species, given how different the hearts of the two species are with respect to size and wall thickness. In Table 1S NMR hearts are approximately 1/3 the mass of mouse hearts, yet in the figure they look the same size. Can you put the magnification of the image. Does the large difference in size contribute to the observed differences in LVDP and heart rates? Fig 6E; What is the biological significance of the significantly larger total adenine nucleotide pool? Fig 6G; Does the significant increase in lactate impact upon pH and is there an indication of better maintenance of acid base balance?

Minor

Introduction

- 3rd paragraph, provide a reference for absence of cardiovascular disease . E.g., Can et al., PMID: 35107705; Delaney PMID: 34424525 and/or others

Methods

- No mention is made as to how you obtained golden moles. How long after their capture hearts were harvested?
- You mention animals were “physiologically age matched”. Looking at Table 1S, this statement is inaccurate, as all mole-rats regardless of their maximum lifespan (~6-37years) were used at ages between 4 and 7 years; “young healthy adults“ would be a better description. Similarly, as mice were ~5 weeks (0.1y) of age this description of them would also be accurate. You mention that many species were wild caught, meaning you cannot accurately know how old they were, but on the basis of size and fur quality you could estimate these to be young, healthy adults. How soon after capture did you harvest the hearts? Could the discrepancy in heart to body weight ratios between wild caught and captive species be explained by time post capture? (Table 1S).
-
- In Table 1S you mention 6 NMRs were used; were 5 of these used in the Langendorff studies or were an additional 5 NMRs used for this study? The correct number should be included in Table 1S. Did you have a control group for the Langendorff studies of hearts only subjected to normoxia to compare baseline glucose, lactate succinate levels. I presume you must have one, given you have the HIF1a under normoxic conditions. I was particularly impressed by the rigor of your western blot studies in that you presented these HIF1a data showing the differing results of the 3 antibodies, but nevertheless all showing the same general trend of increased levels.

Supplemental

- Table 1S : Are these data you yourselves collected for heart weight, body weight, age etc. If so, please say so and if not please provide appropriate references for all the data included in this table. The maximum lifespan of mice on the AnAge website is 4 years. There is no reference “24” in the supplement and ref 24 in the ms refers to succinate efflux.
- You state the age of the B16 mice used in this study is 0.1 years or 4.8 weeks, do they weigh 25g at this age or is that the average species/strain weight for adults? This is important when trying to determine the relative heart weight as a % of total body weight or to evaluate how this differs from the predicted masses from body scaling studies. From my rough estimates the relative heart/body mass % for the mouse heart is 5-fold larger than that of the naked mole-rat and 62 times larger than the largest mole-rat. Can this be simply explained by different energy demands? How do these values compare with the predicted heart masses equations of Schmidt Nielsen (1984; Scaling)? My impression is that the naked mole-rat has a particularly small heart for its body size; although using Schmidt-

Nielsen's equation several of the other bathyergids also have hearts smaller than predicted from the body weight data in Table 1S. Seeing this is novel data it may be worth following up on this too.

Reviewer #2 (Remarks to the Author):

This manuscript describes investigation of the metabolic adaptations exhibited by a rare species, the naked mole rat (NMR), which has apparently evolved to allow survival (possibly even to thrive) in a natural environment of extreme oxygen depletion. The NMR is one of a species within an evolutionary group of African mole-rats, and amongst this group is distinctive for extreme longevity. The investigator achievement in accessing these rare animals to pursue mechanistic studies is impressive. Features of the NMR phenotype relating to hierarchical social structure have been described previously, and some cardiovascular parameters evaluated by others (ie heart rate responses to hypoxic stimuli).

This paper focuses on cardiac metabolic function, which has not been previously explored in detail in the NMR. A range of approaches have been adopted to characterize the cardiac status of the NMR under basal conditions and in a setting of hypoxic challenge with reference to 8 other related species (as close as possible in evolutionary terms) and a standard 'modern' model rodent, the C57/BL6 mouse.

Focussing on the most relevant contemporary contrast experimental model (C57/BL6), in summary:

Under basal conditions the NMR exhibits:

- Lower plasma insulin, glucose, lactate dehydrogenase and triglyceride levels
- Elevated myocardial glycogen content (15-20 x C57/BL6 levels) and glucose-1-phosphate (G-1-P, glycogen metabolite) levels
- Increased creatine accumulation (glycolysis energy reserve) by 44%
- Increased (50%) constitutive expression of HIF-1a during normoxia
- overexpression of the monocarboxylate transporter MCT-1 (plasma membrane cotransport of lactate & H⁺)
- no difference in succinate levels (key TCA intermediate).

In response to ex vivo hypoxia (nuclear magnetic resonance studies, langendorff heart perfusion setting, protocol 20min ischemia, 120min reperfusion), transcriptomic & semi-targeted metabolomic-linked studies demonstrated that the NMR exhibited:

- Marked reduction in myocardial infarct size (near negligible)
- Relatively increased total size of the myocardial adenosine nucleotide pool (sum of ATP+ADP+AMP)
- elevated glucose & lactate levels
- reduced succinate level

Key proteo-metabolic findings include data indicating that:

- the 30 most abundant genes in NMR relate to mitochondrial or ox-phos (vs C57/BL6, which primarily feature genes involved in sarcomeric myofilament structures and oxygen transport)
- the top 10 genes in the NMR (RNA-Seq) are not shared across the other species

Two major mechanistic findings were identified as contributors to the remarkable cardiac resistance to hypoxia:

- a constitutively high expression of HIF1a, which is a recognized key signalling mediator in pathways modulating hypoxic response, and
- reduced mitochondrial ROS production, reflecting a semi-depolarized mitochondrial potential.

1. The findings reported reflect detailed and meticulous work. Notwithstanding, a number of matters arise which could be dealt with by specific author commentary or additional data:

2. The use of terminology 'eusocial' is not very helpful (unusual term). A better description would be more specific, highlighting the animal colobysocial organizational features which confer extreme phenotypic responsive net.

3. In relation to glycogen quantification, why is the C57 mouse myocardial level considered 'undetectable'. This quantification has been reproducibly reported in numerous murine studies. What is the glycogen quantification normalized by and what measurement technique was used? This is not described in the Methods. Provision of non-normalized values would assist in literature benchmarking. Also consideration of canonical and more recently described non-canonical type glycogen flux pathways should be explicitly discussed.

4. The relatively higher level of HIF-1a expression in NMR vs C57 mouse is referred to as 'stabilized'. What does this mean?

5. Is the hypothesis that the longevity of the NMR is due to the lower ROS production – this is implied but not specifically asserted.

6. What specific conclusions or interpretations can be extracted from the dendrogram as presented in Fig 4. More reader guidance is required.

7. For Supp Table 1, the NMR data should be placed at the top of the table for more prominence.

8. The abstract is relatively limited in relation to actual findings reported. More specific reference to the study findings could be reported.

9. It is appreciated that the access to sex-matched experimental groups is limited when dealing with such a rare animal resource. Some commentary about matching (or not) of the other species could be provided. In particular life-span equivalent data from sex matched C57/BL6 animals could be considered.

10. Addition of data relating to fibrosis (or lack of) would enhance the argument relating to ROS level mode of action.

11. More detailed annotations and panel titles within figures would assist readability

12. The importance of the findings reported could be more readily appreciated with inclusion of some relevant mechanistic diagrams. For example the succinate significance in the TCA cycle could be emphasized. Further, the HIF-1a and the ROS-related actions could be depicted in diagram.

Reviewer #3 (Remarks to the Author):

The paper by Faulkes et. al., sets out to investigate the evolutionary adaptations to anoxia and attempts to functionally characterise naked mole-rat's superior protection against ischaemia/reperfusion injury as compared to mouse. The authors first take a comparative approach and perform an extensive RNAseq analysis comparing naked mole-rat gene expression in the heart to 7 other species in the same genera as well as a divergent subterranean rodent and the mouse. The authors also do a more targeted comparative metabolomic analysis with the same species. This dataset is an extremely powerful and valuable resource, which not only supports the authors subsequent mechanistic investigation but more broadly provide highly useful dataset relevant to a wide range of researchers in the field of hypoxia research, cardiovascular disease research, naked mole-rat research, functional genomics research etc. Considering that access to these animals and their tissues is not so simple as well as the table the authors provide to morphology and life history of these animals, I must stress how valuable this dataset is.

The subsequent experiments that were carried out by the authors to gain mechanistic understanding in how the naked mole-rats protect their hearts against ischaemia/reperfusion injury is however, weak in some parts, containing certain gaps, omissions and control flaws that I have pointed out in the comments below. These need to be addressed to strengthen the conclusions made in the manuscript especially those pertaining to the idea that reduced succinate accumulation is responsible for reduced ROS and protection against injury, a mechanism that the authors seem to stress a lot and bring to the forefront in this manuscript. In addition, this 'lowered succinate' phenomena has been described as reverse electron transport, however the authors although elude to this process never overtly describe it. I am curious why RET is never mentioned as it can help put the mechanism brought forward by the authors into a greater context of processes already known about ischaemia/reperfusion in other mammals.

In general also, I think the comprehensive comparative omics analysis provided by the authors could be made use of in a more resourceful way to the study's advantage and support some of the functional observations reported in the second part of the manuscript. For example, why is the link to lactate efflux and elevated MCT transporters never explicitly explored by the authors? Or, is there something in the RNAseq that can explain the lack of succinate accumulation?

I think this study is remarkably valuable and provides very interesting insights to the naked mole-rat's extreme and superior physiology when it comes to dealing with low O₂. With a bit more effort to strengthen the claims, I think this would be a superb addition to our understanding of adapted physiology in hypoxia.

Major comments:

1. (Line 145) Is there a summary about the parameters mentioned here like burrow and soil types, degrees of sociality, lifespan and hypoxia tolerance for each species investigated in this study. This could be added to table S1 and provide comprehensive set of parameters to complement the RNAseq for functionally linking phenotype and genotype via comparative approach.
2. (Line 180) The authors investigate the most highly expressed genes in NMR to be mitochondrial ETC genes. The authors bring this up again in the Discussion. How does this observation relate to previous reports that mitochondrial respiration from isolated heart mitochondria is lower in the naked mole-rat and the activity of the complexes is also lower (eg. the study from Lau et al., 2020 <https://doi.org/10.1016/j.cbpb.2019.110375>).
3. Furthermore, I am not so sure about the implications of looking at the most abundant genes. Even if they are the most expressed genes, is the absolute expression levels between mouse and naked mole-rat different for these genes? Is the protein also more

increased for these genes? In order to make conclusions for very differing heart physiology based on top 30 expressed genes, I would be interested to know how most abundant genes between eg. human and mouse compare? Or perhaps more accessible for these authors, did the authors perform 30 most abundant gene analysis between the other African mole rats and NMR, or versus other African mole rats within themselves and against mouse.

4. (Line 220) The authors measure plasma insulin levels but do not include how they measured insulin in Materials and Methods. Naked mole-rat insulin has mutation effecting the beta-sheet and is notoriously resistant to measurement by mouse and human ELISAs. If the authors used mouse or human kits, the much lower insulin level could be due to detection efficiency as ELISAs are antibody based and therefore would not detect the naked mole-rat version of insulin. Please check, elaborate and validate that the insulin is really detected with similar efficiency.

5. Furthermore, I did not find in the materials and methods a description for how any of the 4 plasma metabolites were measured. Please include details of how glucose, insulin, triglycerides and lactate dehydrogenase were measured.

6. (Line 232) Although showing slightly differing results, there has already been a publication on HIF1alpha protein as well as mRNA expression in the naked mole-rat compared to mouse in the heart. The strain of mouse used was different perhaps explaining the difference in observations. The authors should at least cite this paper and omit the word unexpected since stabilised HIF1alpha protein expression has already been shown in the study by Xiao et al, 2017. <https://doi.org/10.18632/oncotarget.22767>

7. (Line 241) The LDH1 and MCT1 blots shown in Suppl Fig 1 showing the expression of Hif1alpha targets are consequently overexpressed has a few issues:

a. Although stated that MCT1 is overexpressed the representative blot and the band density quantification (Suppl Fig 1D) shows non-significance.

b. Why do the authors use different housekeeping proteins in every blot that they do? In any case it is questionable whether the use of housekeeping/loading controls makes sense in comparing across species or rather adds a confounding factor since it seems like none of the housekeeping proteins chosen actin (Figure S1a), Vinculin (S1c) and Tubulin (S1E) have consistent expression between mouse and naked mole-rat. They all seem to have species specific differential protein expression and therefore are not appropriate loading controls. If the proteins were loaded at the same concentrations, perhaps foregoing the normalisation based on housekeeping protein and show equal loading based on a whole protein staining like Ponceau is more appropriate for across species comparison.

c. The most direct consequence of HIF1alpha stabilisation would be the RNA expression of HIF1alpha targets rather than protein expression. In order to make the point that HIF1alpha targets are upregulated in the naked mole-rat, mRNA for target genes should be quantified for known targets like VEGF, GLUT1, LDH and MCT1. This could already be ascertained from the RNAseq data (mouse versus NMR) the authors performed but did not include in the supplementary data set.

8. (Line 255) The authors use the word normothermic ischaemia. Please define the exact temperatures used as the general reader is not aware of normothermic values for naked mole-rats, especially as their thermoneutral zone is a wide range of 28-32 degrees.

9. The influence of temperature may impact the extent of injury of the heart. Therefore the mouse undergoing normothermic ischaemia at 37 degrees could sustain greater damage purely due to elevated temperatures compared to the naked mole-rat undergoing normothermic ischaemia at 32 degrees or less. It is well known that reducing temperature during ischaemia protects the tissue at risk. I understand naked mole-rats may be precious and limiting, so to control for the significant confounding factor, the authors should perform the ischaemia/reperfusion protocol on mouse hearts under the same temperature conditions used for the naked mole-rat and assess the injury thereafter. Likewise, the metabolites

measured during normothermic ischaemia may be a result of slower metabolism due purely to lower temperatures. Metabolites, like succinate, other TCA cycle intermediates and lactate should also be measured in mouse ischaemic hearts under temperature conditions used for the naked mole-rats.

10. Figure 4C shows a representative slice from the heart. Since the naked mole-rat hearts are different sizes and the injury zone can be located in slightly different sites, can the authors provide images of the entire heart, i.e. all 4-5 slices and elaborate whether injury size in figure 6D was quantified from a distinct slice or the entire heart?

11. (Line 255) The authors state that the NMRs have enhanced energy supply by showing total adenine pool. Why are the authors only reporting total adenine pools? From Figure 4a, it seems like the authors are able to measure at least separate ATP and AMP pools. The ratio of ATP to ADP or AMP would be a more useful and insightful measure of whether the naked mole-rats actually maintain higher energy supplies. If the ¹H-NMR analysis performed allows separation of adenine pools, please reanalyse and show these separately.

12. (Line 257) The authors report reduced succinate accumulation, however they do not provide a baseline measurement of succinate levels of ex vivo perfused hearts prior to ischaemia. Although there is succinate measurement from freshly dissected hearts from mouse and naked mole-rat in Figure 4f which can hint at baseline levels, we cannot conclude that the same amounts exist in isolated perfused hearts at the end of 20 minute equilibration period. The authors should provide metabolite measurements for naked mole-rats and mice either after the equilibration period just prior to ischaemia exposure or perhaps as a better control after 40 minutes of normoxia perfusion (20 minutes equilibration and 20 minutes normoxia versus 20 minutes ischaemia).

Furthermore, the units for Succinate in figure 4F and Fig 6H are in different Units (normalised peak area versus arbitrary units). Can the y-axis be made consistent between different figures for the metabolites.

13. Alternatively, if access to naked mole-rats may be a problem, but optimally in addition to the experiment in point 12, evidence of succinate accumulation in ischaemia should be further supported by looking at other TCA cycle intermediates. In another publications by some of the same authors of the current article (Couchani 2014), citrate, aconitate, alpha-ketoglutarate, fumarate and malate have been shown to be unchanged under ischaemia relative to normoxic conditions whereas succinate accumulates 12-fold. Therefore, the authors should put in effort to show succinate in relation to the other TCA cycle intermediates and their changes relative to normoxic conditions to really pinpoint the lack of succinate accumulation and not just a reduced overall mitochondrial metabolism in the naked mole-rat.

14. The authors claim that it's succinate accumulation which leads to ROS and IR injury and that by having reduced succinate accumulation under ischaemia, the naked mole-rats are therefore protected against injury. ROS however has not been measured in the naked mole-rat under IR conditions in order to make the link. Therefore, the conclusion the authors make here based on weakly controlled succinate accumulation experiment and no ROS measurements is unsubstantial this point. To strengthen this conclusion, the authors should measure ROS in baseline and ischaemic/reperfused tissue and compare accumulation of ROS between naked mole-rat and mouse.

15. The authors claim that naked mole-rats accumulate lactate which then gets rapidly effluxed in the first minute of reperfusion. The authors don't go so far as to overtly say this may additionally be a protective mechanism, but by showing this piece of data imply it. However, the presented data is incomplete in its current state to make any reasonable conclusion about whether naked mole-rat have greater efflux of lactate and therefore lactate's role in IR injury. The authors can strengthen this very important observation with the following experiments.

a. The authors first claim that naked mole-rats accumulate lactate during ischaemic phase (Fig6G). The authors however only show lactate levels in ischaemic phase and it therefore cannot be concluded with certainty that the lactate has indeed accumulated compared to baseline levels. This can be amended by the experiment I suggested in Point 12.

b. The authors show that at baseline, the lactate in the effluent is the same in mouse and naked mole-rat (Suppl. Fig 6a). However, the measurements for lactate after reperfusion (Suppl. Fig 6b) is only shown for the naked mole-rat. Why don't the authors show lactate in the effluent for the mouse as well? Without comparing lactate in effluent after reperfusion to the mouse, we cannot draw any conclusion from the dataset. As such, we do not know whether a species, like mouse, which is not protected from ischaemia, also effluxes lactate at reperfusion and restores normal levels quickly or whether this is indeed what differentiates the naked mole-rat from mouse and is therefore unique to the naked mole-rat hinting that it may be contributing to the resistance to IR injury in this species.

Generally, how naked mole-rats deal with lactate accumulation during ischaemia or at reperfusion thus avoiding the problem of lactic acidosis is a highly fascinating topic and the authors here can uncover a potentially interesting phenomena that can contribute to our understanding of protective heart physiology under ischaemia. Therefore, providing evidence that the naked mole-rat has a greater capacity for efflux could be extremely strengthening for this manuscript. If the dataset is completed as I suggest above, this figure can come out of supplementary and into the main manuscript.

16. (Line 405) In methods section, please provide details to the temperature used for Langerdorff experiments for each species

17. Supplementary Dataset 2 provides complete dataset for all genes for NMR versus all other African mole rat species. Why is the dataset for naked mole-rat versus mouse not provided especially since the rest of the manuscript focuses on mouse against naked mole-rat comparison? I think including this dataset would complement the rest of the study.

Minor comments

1. Line 142 Mahali mole rat needs a hyphen for consistency

2. The Authors claim in the legend for Fig 6 that n=5 is used for ascertaining the infarct size, when in fact from Figure 6D only 2 points are visible for naked mole-rat and 3 points are visible for the mouse.

3. Paragraph beginning with Line 307 should cite the Xiao et al, 2017 paper as well.

4. (Line 352) claims that the study provides insights into the NMR cardiac adaptations to the dark and O₂. It is clear that the manuscript handles adaptation to low oxygen but the significance of this finding to the dark is less clear, especially as the holding conditions for NMRs described has a light/dark cycle. Please omit or elaborate.

Reviewer #4 (Remarks to the Author):

In this study RNAseq and metabolomics have been performed on the cardiac tissues of naked mole-rats and other mole-rat genera, providing fascinating genetic and metabolomic insight into the unique characteristics of the naked mole rat. I do however have a number of comments to be addressed, primarily surrounding the metabolomics methodology.

1. A bit more detail is needed on the extraction, storage and processing of the heart samples to demonstrate sample integrity was maintained for downstream analysis. Some sections of

the method indicate naked mole rat and other mole rat samples were not subjected to the same analysis (e.g. the cardiac perfusion). It is not clear exactly what processes the samples went through prior to analysis and whether they were in fact treated exactly the same, so please clarify this. Any variation in sample treatment could have significant effects on metabolites, creating artificial differences.

2. The authors state that NMR analysis was “semi-targeted”. Can the authors expand on this to describe how they selected the metabolites discussed? Were specific analytes the focus of this study from the start or was some kind of statistical framework used to select analytes to identify and compare?

3. Were there any statistical differences in other metabolites that were either not discussed or not identified? Although not discussed, this could be useful and interesting information to share.

4. How many metabolites were used in the LDA and PCA plots - all detected features or just the metabolites listed on the heatmap in Fig 4A? Please specify in the legend or methods.

5. Regarding the LDA plot in figure 4B, presumably the ellipses depict a confidence interval. If so, this should be detailed in the figure caption.

Faulkes, Eykyn, Miljkovic et al . NCOMMS-23-27279-T

"Naked mole rats have distinctive cardiometabolic and genetic adaptations to their underground low-oxygen lifestyles"

We would like to thank the Reviewers for the detailed review of our Manuscript and their critical feedback. We are thrilled to see they found our work ***meticulous, original, novel, our dataset valuable*** and recognising that our manuscript provides ***very interesting insights*** to the naked mole-rat's extreme and superior physiology.

We would also like to thank them for commending our effort to access these difficult to reach, non-model organisms which indeed was a challenging task compounded by the COVID-19 pandemic during which most of the studies presented here took place.

We have taken their comments on board, performed additional experiments and incorporated suggested changes ***thus altering the manuscript, figures (additional/new Figure 7 and Figure 8), and supplement extensively*** (MS edits in red font). Point-by point responses to Reviewers' comments are provided below:

Reviewer #1 (Remarks to the Author):

1. This well written manuscript addresses differences in bulk RNAseq of hearts from 9 species of subterranean mammals comparing each of these data sets with that of the naked mole-rat [NMR] and highlights key differences that may contribute to the extraordinary tolerance of NMRs to hypoxic conditions and ischemia /reperfusion [I/R] injury. The authors then follow up with an ex vivo study examining the responses of mice and mole-rats to I/R that further confirms NMRs are indeed more resistant than mice to I/R. These are original novel data that would be gladly received by all working on the comparative biology of subterranean animals as well as those interested in possibly translating these findings into biomedical application relevant to cardiac diseases.

We would like to thank the Reviewer for their kind comments and finding our manuscript well-written, data novel as well as translationally relevant.

2. With so large a data set it is understandable that reported findings have been specifically chosen, however their well- developed introduction suggests that something more than what was discussed will be included and further justifies this: For example, surprisingly looking at the volcano plots (Fig 3), the NMR shows the greatest similarity to that of the golden mole! Yet this is not addressed.

We thank the Reviewer for their comment. However, we do not agree that the volcano plots in Fig 3 indicate a strong similarity between NMR and Golden mole. This is because the most differentially expressed genes are 2[^]10 fold upregulated in NMR vs golden mole. While these fold differences between NMR and golden mole may be slightly lower than those between NMR and other groups, they are still starkly different. Moreover, the patterns of which specific genes are different are similar between the NMR and all the other subterranean species as is shown in Figure 3. Similar fold changes are found between NMR and Cape mole-rat so we do not believe that these data can be used to infer greater similarities between the NMR and the golden mole.

3. Equally surprising is the fact that PCA analysis suggests greater similarity amongst the southern African Bathyergids regardless of whether or not they are eusocial, solitary and/or social and therefore subject to different degrees of hypoxia stress at least in the nest area where multiple animals versus a single animal may be resting. Despite including information regarding these features in Fig 1, the impact of their divergent lifestyles on hypoxia tolerance is not discussed.

The PCA analysis of both the RNAseq and metabolomics data does seem to point to a high degree of similarity between the other genera of mole rats compared to NMR, regardless of whether they are eusocial, solitary and/or social. This suggests that the other southern African Bathyergids have not undergone such divergent evolution from their common ancestor as is the case for NMRs. It also suggests that the range of evolutionary pressures imposed on these related species have not been as great as for the NMR. This is probably due to the quite extreme lifestyle with regard to hypoxia that is experienced by the NMRs that has a disproportionate effect on their evolutionary divergence, compared to the differences caused by different social structures, for whatever reasons - despite group living and multiple occupancy of nest chambers in the other southern African Bathyergids. We have commented briefly on this in the text on page 3 of the manuscript (Introduction, Paragraph 3).

4. Nor do the authors discuss the results of how gene profiles of their outgroup and that of mice differs from the various mole-rat species, despite the fact that they highlight the importance of the need for an outgroup in the introduction.

The information on gene profiles suggested for inclusion regarding the other mole-rats, golden mole and the mouse was actually included in supplementary Data file S2, which shows the 30 most expressed genes (normalised as FPKM) across all the sample sets. Within this data file table cells are coloured to show shared expressed top 30 genes compared to the NMR.

Unique NMR genes are highlighted in yellow and a brief description was given in the text on page 5 as follows: “Comparison of the top 30 expressed cardiac genes across all the species analysed exemplifies the unique gene expression pattern of NMRs. Only 13 genes listed in the NMR top 30 are found in the top 30 of the other species, and the top 10 in the NMR are not shared across the other species (Supplementary Information Data S1)”. However, we did not make explicit the comparison with the out group or with mice.

We have reworded the above text in our manuscript to clarify this point, which now reads “**Comparison of the top 30 expressed cardiac genes across all the species analysed, exemplifies the unique gene expression pattern of NMRs and their divergent phenotype when compared to other mole-rats and the mouse and golden mole outgroups. The top 10 expressed genes in the NMR do not appear within the top 30 from any other species while many of the top 10 genes from the other species are shared in common (Supplementary Information Data S2)**” (page 5).

Furthermore, the dendrogram in Figure 2A (showing hierarchical clustering of samples by similarity based on the expression level of all genes) clearly emphasises similarities among the non-NMR mole-rats and the golden mole (they are all in a clade together, rather than the golden mole forming an outgroup lineage) - with NMRs as a distinct reciprocally monophyletic clade.

It would have been nice to see their data on mice gene profiles included in Figure 2B.

We agree! However, we found this challenging to do without distorting the current way in which the data are displayed. We did spend quite some time trying to include the mouse data with the other subterranean species, but the analysis pipeline would not run effectively, largely due to annotation problems mapping the mouse to the NMR genome and *vice versa*. We discussed this in detail with experienced colleagues in the BGI Genomics (company that performed RNA sequencing and analysis) bioinformatics team however, even after trying several different approaches they were unable to perform the analysis in a satisfactory way. We have extended the note to this effect in the Methods section (page 11).

5. If RNAseq was not done on mouse hearts in the absence of I/R, there are numerous other studies where this has been done that maybe could be included to show just how different mice and NMR heart gene profiles are.

RNA seq was done on mouse hearts in the absence of ischemia/reperfusion and the data set is included in Supplementary Information Data S2. However, unfortunately for the reason outlined above (under point 4) direct comparison of the mouse with NMR was limited/not possible.

6. Could the unique transcriptomic and metabolomic profile of the naked mole-rat infer that the NMR is taxonomically distinct as suggested by Patterson 2014. <https://doi.org/10.1111/zoj.12201> [18] ? Are there potentially other explanations as to why this species is such an outlier.

Indeed yes, we agree that the NMR is taxonomically distinct. All molecular phylogenetic studies to date have shown that the NMR is the first (extant) lineage to diverge from the common ancestor within the clade of related other southern African Bathyergids, possibly around 30-35 million years ago. As such NMR has had a long evolutionary time to become highly specialised within the subterranean niche and it has always been taxonomically distinct (a monotypic species in the genus *Heterocephalus*). Over the years of research on the NMR virtually all studies have revealed it to be an outlier within the African mole-rat clade as a result (presumably) of its long evolutionary history. Patterson’s paper suggests putting the NMR into its own family - however even if this is done, this does not alter the monophyletic nature of the African mole-rat clade as a whole as they all share a single common ancestor no matter how you name the lineages. This issue is reviewed in our citation 1: Buffenstein, R. et al. The naked truth: a comprehensive clarification and classification of current 'myths' in naked mole-rat biology. *Biol Rev Camb Philos Soc* 97, 115-140, doi:10.1111/brv.12791 (2022).

We have drawn further attention to this issue in our text as follows: “**Comparison of the top 30 expressed cardiac genes across all the species analysed exemplifies the unique gene expression pattern of NMRs and their divergent phenotype when compared to other mole-rats**” (page 5).

7. The second section of the paper uses ex vivo preparations of naked mole-rat and mouse hearts and subjects these to ischemia /reperfusion using standard techniques. The following sentence “Ischemia/reperfusion outcome in NMRs was compared to C57Bl6 mouse heart due to their anatomical, physiological, and genetic similarity to humans”, in their methods on Langendorff preparations seems spurious. The data suggest the NMR is very different to C57BL/6 mice in terms of heart size, function and gene profile so are you simply saying you use the mouse as a proxy for humans? Given that both NMRs and humans have far lower heart rates and more similar ejection fractions, this justification seems wrong and is not needed for it is perfectly acceptable to compare the mouse and the NMR as similar sized small rodents with widely different tolerance of hypoxia. The I/R results obtained are terrific and strongly support the findings from the gene profile data.

We are pleased that the Reviewer finds our I/R results terrific and in strong support of our genetic profile data. We also agree with the Reviewer that it is acceptable to compare NMR with the C57/BL6 mouse in this context. We thank the reviewer for their observation and comment. We have now added your statement about the validity of comparing these species to our Methods (page 12).

In our experiments mice were used as hypoxia non-resistant control group because they are widely used organisms in cardiovascular physiology pre-clinical and mechanistic studies to replace the use of primates and humans (PMID: [23829104](https://pubmed.ncbi.nlm.nih.gov/23829104/), <https://www.jax.org/why-the-mouse/excellent-models#:~:text=Almost%20all%20of%20the%20genes,digestive%2C%20hormonal%20and%20nervous%20systems>). Whilst mice and humans are morphologically diverse, both species are mammals and have many biological similarities in cardiac physiology and pathology. We have now added a reference for our statement (Reference 37).

8. Figure 6C: I would have liked to see representative cardiac section images from control (normoxia) mice and mole-rats rather than only images post I/R, showing an "infarction". This MI is not labelled on the figure, so to the uninitiated they just look like lighter images in mice. Having a control group set of images would also enable a better understanding of the changes with I/R treatment in each species, given how different the hearts of the two species are with respect to size and wall thickness. In Table 1S NMR hearts are approximately 1/3 the mass of mouse hearts, yet in the figure they look the same size. Can you put the magnification of the image.

We have taken the Reviewer's comment on board and re-made the figure to also include a scale bar for reference (Figure 6C) and we have also now used clearer images. The reviewer is correct- NMR hearts are smaller than those in mice and we had originally adjusted the image to make them comparable in size to C57/BL6 mouse hearts for presentation purposes. We apologise that this was misleading. We have now amended the figure and included this corrected version in new Figure 6C. We have also added additional information to the figure legend stating that white tissue is the ischemic scar whilst the red tissue is viable healthy myocardium.

0.3cm

C57/BL6

NMR

Due to a number of technical issues (for example, when the coverslide is pressed against heart sections for scanning it squashes and enlarges the section), cross sectional images do not reflect the true heart size as the tissue sections are compressed prior to scanning. Tissue sections prepared using frozen hearts and a Zivic mouse heart sectioning gauge produces thicker sections compared to paraffin embedding/sectioning thin sections.

9. Does the large difference in size contribute to the observed differences in LVDP and heart rates?

Heart size alone in NMRs is not the determinant of their unusual cardiac physiological stress response. For example, they also have slow basal metabolic rate and lower body temperature. *In vivo* functional measurements have shown that NMRs have lower heart function at baseline and that the relatively low basal cardiac function and enhanced cardiac reserve of NMRs are likely to be ecophysiological adaptations to life in an energetically taxing environment (PMID: 24363308, PMID: 22367435). We have now added this comment to our manuscript (page 6, ref 15).

10. Fig 6E; What is the biological significance of the significantly larger total adenine nucleotide pool? Fig 6G;

We are sorry for not making this clearer in our discussion. Larger total adenine nucleotide pool post-ischemia signifies increased readily available energy (ATP) reserve at the time of metabolic and physiological stress. We have now amended our manuscript to add this comment (page 7).

11. Does the significant increase in lactate impact upon pH and is there an indication of better maintenance of acid base balance?

The Reviewer raises an important aspect of NMR metabolism. Whilst we could not perform ^{31}P NMR spectroscopy experiments to study pH regulation in perfused NMR hearts, we have performed additional experiment (new Figure 7) on the coronary effluent from perfused NMR hearts that sheds light on the way NMR hearts handle lactate and thus manage their lactic acidosis/intracellular pH post I/R. These additional experiments enabled us to now make direct comparison between lactate production and efflux in normoxic perfusion, immediately after ischemia (1 min reperfusion) and 1 hour into reperfusion (NMR vs C57/BL6 mouse hearts). Our findings are now summarised in our new Figure 7A in the main manuscript (lactate efflux, MCT1 and LDH expression).

New Figure 7

We find that NMR hearts efflux less lactate at baseline in normoxia as well as at 1min and 1h post-reperfusion, compared to C57/BL6 mouse (Figure 7A). However, during ischemia NMR hearts have elevated intracellular lactate compared to mice (Figure 6G) likely due to increased availability of carbohydrate reserves (glycogen, Figure 4C) which is broken down to yield increased endogenous glucose pool (Figure 6F) and increased endogenous lactate arising from glycolysis (Figure 6G). NMR hearts display increased MCT-1 expression (Figure 7B) and elevated myocardial LDH expression (Figure 7C), therefore the observation that NMR hearts efflux less lactate than C57/BL6 hearts during reperfusion (Figure 7A) is paradoxical. MCT-1 and LDH are also expressed in the mitochondria and therefore endogenous lactate could also be converted to pyruvate and used as a metabolic fuel rather than being effluxed as metabolic waste. This paragraph has been added to our manuscript (page 6)

Minor

1.Introduction

• 3rd paragraph, provide a reference for absence of cardiovascular disease . E.g., Can et al., PMID: 35107705; Delaney PMID: 34424525 and/or others

We thank the Reviewer and we have now inserted (refs 15-18).

2. Methods

- No mention is made as to how you obtained golden moles. How long after their capture hearts were harvested?
- You mention animals were “physiologically age matched”. Looking at Table 1S, this statement is inaccurate, as all mole-rats regardless of their maximum lifespan (~6-37years) were used at ages between 4 and 7 years; “young healthy adults“ would be a better description.

Thank you for this point, we have now added this correction to our manuscript. Similarly, as mice were ~5 weeks (0.1y) of age, this description of them would also be accurate.

You mention that many species were wild caught, meaning you cannot accurately know how old they were, but on the basis of size and fur quality you could estimate these to be young, healthy adults. How soon after capture did you harvest the hearts? Could the discrepancy in heart to body weight ratios between wild caught and captive species be explained by time post capture? (Table 1S).

We thank the Reviewer for these points and we agree that the discrepancy in heart to body weight ratios between wild caught and captive species can be explained by time post-capture. All wild-caught African mole-rat species were maintained in captivity for at least one year before tissue harvesting and we have now added this comment to our Methods (Animals section, page 10). A noticeable drop in the heart-to-body mass ratio is observed in African mole-rats after a year in captivity as shown in the summary Table below.

Species	1-year captivity Heart-to-body mass (mg/g) [this study]	1- 3 week captivity Heart-to-body mass (mg/g) [new data]
Georchus capensis	0,51	4,46
Bathyergus suillus	0,14	3,04
Cryptomys hottentotus hottentotus	0,98	4,13
Cryptomys hottentotus pretoriae	0,80	3,73
Cryptomys hottentotus mahali	0,75	4,93
Cryptomys hottentotus natalensis	0,74	
Fukomys damarensis	0,75	
Heterocephalus glaber	1,45	
Amblysomus hottentotus	1,30	
Mus musculus	6,21	

Table 1. Impact of captivity on heart to body mass ratio

The Golden moles were kept captive for approximately 3 weeks due to their short life-span and difficulty maintaining this species in captivity. We have now added this comment to our Methods-Animals section (page 9).

3. In Table 1S you mention 6 NMRs were used; were 5 of these used in the Langendorff studies or were an additional 5 NMRs used for this study? The correct number should be included in Table 1S.

We thank the Reviewer for spotting this. We have updated the numbers of naked mole rats used in our study overall.

4. Did you have a control group for the Langendorff studies of hearts only subjected to normoxia to compare baseline glucose, lactate succinate levels. I presume you must have one, given you have the HIF1a under normoxic conditions.

We have previously published Langendorff perfused control (C57/BL6) mouse heart metabolomic data (PMID: 32766828). Due to the very limited access to NMRs compared to mice, we are unable to cull the animals to measure cardiac metabolism in normoxic Langendorff model, however we have already presented NMR cardiac metabolism in normoxia in Figure 4. Furthermore, we published NMR metabolism in normoxia in our previous manuscript (PMID: 31771414). Therefore, we feel that normoxic experiments of an NMR Langendorff heart would not add additional insight to our study.

5. I was particularly impressed by the rigor of your western blot studies in that you presented these HIF1a data showing the differing results of the 3 antibodies, but nevertheless all showing the same general trend of increased levels.

We thank the Reviewer for noting our efforts to probe for multiple HIF1alpha epitopes.

6. Supplemental

• Table 1S : Are these data you yourselves collected for heart weight, body weight, age etc. If so, please say so and if not please provide appropriate references for all the data included in this table.

We thank the Reviewer for noting this. Data has been collected by the study authors and using the animals from the analysis cohorts. We have now added a sentence to our Table S1 legend in order to clarify this.

7. The maximum lifespan of mice on the AnAge website is 4 years.

We apologise for this lapse, which has now been amended in Supplementary Table 1.

8. There is no reference “24” in the supplement and ref 24 in the ms refers to succinate efflux.

We apologise for this lapse which has now been amended and reference 1 of Supplement included.

9. You state the age of the B16 mice used in this study is 0.1 years or 4.8 weeks, do they weigh 25g at this age or is that the average species/strain weight for adults?

This is the average weight of mice used- should correspond to ~7 weeks of age as per Charles River growth curves, which was where the mice were obtained from

(https://www.criver.com/sites/default/files/resources/doc_a/C57BL6MouseModelInformationSheet.pdf).

7.5 week old mice correspond to 0.1 years (0.1342).

10. This is important when trying to determine the relative heart weight as a % of total body weight or to evaluate how this differs from the predicted masses from body scaling studies. From my rough estimates the relative heart/body mass % for the mouse heart is 5-fold larger than that of the naked mole-rat and 62 times larger than the largest mole-rat. Can this be simply explained by different energy demands? How do these values compare with the predicted heart masses equations of Schmidt Nielsen (1984; Scaling)? My impression is that the naked mole-rat has a particularly small heart for its body size; although using Schmidt-Nielsen's equation several of the other bathyergids also have hearts smaller than predicted from the body weight data in Table 1S. Seeing this is novel data it may be worth following up on this too.

The reviewer raises a very interesting point. Notwithstanding the effects of captivity on heart/body weight ratios outlined in Point 2 above, the discrepancy between mole-rat heart sizes and Schmidt-Nielsen's equation certainly warrants further investigation. It is likely that (as the reviewer mentions) different energy demands play a part in this but there are also a number of other variables. This very topic is indeed part of another ongoing study by authors on this study (Hart and Bennett) for which a separate manuscript by them is planned. Further consideration of this important point is therefore beyond the scope of this current manuscript.

Reviewer #2 (Remarks to the Author):

This manuscript describes investigation of the metabolic adaptations exhibited by a rare species, the naked mole rat (NMR), which has apparently evolved to allow survival (possibly even to thrive) in a natural environment of extreme oxygen depletion. The NMR is one of a species within an evolutionary group of African mole-rats, and amongst this group is distinctive for extreme longevity. The investigator achievement in accessing these rare animals to pursue mechanistic studies is impressive. Features of the NMR phenotype relating to hierarchical social structure have been described previously, and some cardiovascular parameters evaluated by others (ie heart rate responses to hypoxic stimuli).

We thank the Reviewer for kind feedback on our efforts to design and execute the study using these rare and difficult to access species which indeed was a challenging task compounded by the COVID-19 pandemic during which most of the studies presented here took place.

This paper focuses on cardiac metabolic function, which has not been previously explored in detail in the NMR. A range of approaches have been adopted to characterize the cardiac status of the NMR under basal conditions and in a setting of hypoxic challenge with reference to 8 other related species (as close as possible in evolutionary terms) and a standard 'modern' model rodent, the C57/BL6 mouse.

Focussing on the most relevant contemporary contrast experimental model (C57/BL6), in summary:

Under basal conditions the NMR exhibits:

- Lower plasma insulin, glucose, lactate dehydrogenase and triglyceride levels
- Elevated myocardial glycogen content (15-20 x C57/BL6 levels) and glucose-1-phosphate (G-1-P, glycogen metabolite) levels
- Increased creatine accumulation (glycolysis energy reserve) by 44%
- Increased (50%) constitutive expression of HIF-1a during normoxia
- overexpression of the monocarboxylate transporter MCT-1 (plasma membrane cotransport of lactate & H⁺)
- no difference in succinate levels (key TCA intermediate).

In response to ex vivo hypoxia (nuclear magnetic resonance studies, langendorff heart perfusion setting, protocol 20min ischemia, 120min reperfusion), transcriptomic & semi-targeted metabolomic-linked studies demonstrated that the NMR exhibited:

- Marked reduction in myocardial infarct size (near negligible)
- Relatively increased total size of the myocardial adenosine nucleotide pool (sum of ATP+ADP+AMP)
- elevated glucose & lactate levels
- reduced succinate level

Key proteo-metabolic findings include data indicating that:

- the 30 most abundant genes in NMR relate to mitochondrial or ox-phos (vs C57/BL6, which primarily feature genes involved in sarcomeric myofilament structures and oxygen transport)
- the top 10 genes in the NMR (RNA-Seq) are not shared across the other species

Two major mechanistic findings were identified as contributors to the remarkable cardiac resistance to hypoxia:

- a constitutively high expression of HIF1a, which is a recognized key signalling mediator in pathways modulating hypoxic response, and
- reduced mitochondrial ROS production, reflecting a semi-depolarized mitochondrial potential.

1. The findings reported reflect detailed and meticulous work. Notwithstanding, a number of matters arise which could be dealt with by specific author commentary or additional data:

2. The use of terminology 'eusocial' is not very helpful (unusual term). A better description would be more specific, highlighting the animal colobysocial organizational features which confer extreme phenotypic responsive net.

We think that the reviewer's comments may have been autocorrected to "colobysocial" and "phenotypic responsive net". However, selection of terms for describing sociality is a highly relevant point to raise. The term eusocial is commonly used to describe certain social insect societies and from 1981 also entered the mammal sociobiology literature when NMRs were first described as a eusocial mammal (Jarvis, 1981) as they fulfil a specific definition (reproductive division of labour, overlapping generations and cooperative care of young). Since 1981 there has been much discussion in the literature regarding definitions of sociality more generally (there is no "one size fits all answer" as to how to define levels of sociality). These issues in relation to mole-rats together with justification and discussion of the use of the term eusocial are reviewed in our reference [1] cited at the end of the first sentence of the introduction (Buffenstein, R. *et al.* The naked truth: a comprehensive clarification and classification of current 'myths' in naked mole-rat biology. *Biol Rev Camb Philos Soc* **97**, 115-140, doi:10.1111/brv.12791 (2022).

3. In relation to glycogen quantification, why is the C57 mouse myocardial level considered 'undetectable'. This quantification has been reproducibly reported in numerous murine studies. What is the glycogen quantification normalized by and what measurement technique was used? This is not described in the Methods. Provision of non-normalized values would assist in literature benchmarking.

We thank the Reviewer for this point. Quantification of glycogen was performed by ¹H magnetic resonance. ¹H MRS gives a measure of the concentration of glucose monomers that are present in the observed peak normalized to the reference trimethylsilyl propanoic acid (TSP) peak. Being a large macromolecule, with possible differences in the mobility of glycosyl units, glycogen has been reported to be fully visible by MRS. However, we used a modified dual-phase Folch extraction method used for separating aqueous and lipid metabolites which has not been optimized for the extraction of glycogen per se. A direct comparison of the glycogen tissue values obtained in this study to the values published by traditional enzymatic techniques will be at variance. Notwithstanding, the application of the same methods to all species allows an assessment of relative changes between groups and therefore our assessment of between species variation remains valid. We considered this point in our previous paper (PMID: 31771414) and have added a statement in the methods for the current study (Page 13). All of our raw, non-normalized metabolomic data is deposited in the dryad library linked to our manuscript.

4. Also consideration of canonical and more recently described non-canonical type glycogen flux pathways should be explicitly discussed.

We thank the Reviewer for this valid suggestion about this omission on our part. We have now discussed in our Discussion some recently discovered aspects of the regulatory role of glycogen in cardiac physiology (References 21-26).

4. The relatively higher level of HIF-1 α expression in NMR vs C57 mouse is referred to as 'stabilized'. What does this mean?

We apologise for not making this clearer. Stabilised is a widely used term in HIF-1 α literature, as normally the HIF-1 α protein is rapidly degraded in the presence of oxygen by proteases. In some circumstances this degradation process is prevented, so the protein is "stabilised" meaning it is non-degraded. We have now inserted the term non-degraded in brackets where stabilised has been used (Page 4, Page 6).

5. Is the hypothesis that the longevity of the NMR is due to the lower ROS production – this is implied but not specifically asserted.

We thank the Reviewer for this question. Given that we have not assessed ROS production in NMR hearts in our study, it is difficult to comment on this aspect of their cardiac biology. However, several other studies have investigated the production of free radicals and oxidative damages in the naked mole rat, and the results are both puzzling and conflicting showing increase, decrease and no change thus it is not possible to explicitly state the role of ROS in our manuscript. (i.e. PMID: 2173654, PMID: 21736541, PMID: 17596208, PMID: 21736541 to name a few). We have also altered the text on page 9 to remove the implication of ROS involvement in longevity.

6. What specific conclusions or interpretations can be extracted from the dendrogram as presented in Fig 4. More reader guidance is required.

The dendrograms in Fig 4 (and also in Fig 2) show clusters of data joined by branches and nodes according to their degree of similarity. This is a standard presentation of hierarchical cluster analysis and we do not feel it requires further explanation in the text as it is a standard output from Matlab, and widely used in the literature. But for clarification the tree diagram on the top x-axis of Fig 4 shows clustering of the genotypes where NMR fall into the same cluster, therefore being most similar to each other, while they are dissimilar to the other genotype which fall in different trees. Similarly clustering of metabolites on the left y-axis shows metabolites that are related because they are in the same pathway or influenced by a common biological process. For interest alanine, lactate and succinate all fall in the same cluster.

7. For Supp Table 1, the NMR data should be placed at the top of the table for more prominence.

We thank the Reviewer for this suggestion and we have now altered the Supplementary Table 1.

8. The abstract is relatively limited in relation to actual findings reported. More specific reference to the study findings could be reported.

We have altered the Abstract to make more specific reference to the study findings.

9. It is appreciated that the access to sex-matched experimental groups is limited when dealing with such a rare animal resource. Some commentary about matching (or not) of the other species could be provided. In particular life-span equivalent data from sex matched C57/BL6 animals could be considered.

A column in Supplementary Table 1 shows the sex ratios of each mole-rat group used as both sexes were used throughout (including NMR; n=7 males n=7 females). We have now added a line to our methods to clarify this point (Animals section page, 9). The C57/BL6 mice used were male as is stated in Table 1S. They were young adults, chosen to match the mole-rat species as stated in our Methods,

Animals section, paragraph 1 (page 9). Only male mice were used because it was a small cohort of mice and we wished to avoid introducing variability as that would require culling more NMRs to power the analyses. We agree with the Reviewer that in future NMR studies we should include both male and female cohorts of C57/BL6 mice.

10. Addition of data relating to fibrosis (or lack of) would enhance the argument relating to ROS level mode of action.

There is no direct correlation between ROS production and NMR biology including longevity and preservation of cardiac function in the NMR studies published to date (this is also outlined in point 5 above). NMRs only exhibit ventricular fibrosis from 13-24 years (PMID: 24906918), which is much older than the NMRs we used. Furthermore, even changes in ventricular fibrosis with age in NMRs do not impact cardiac function which is maintained for at least 75% of their maximum life span potential (40 years, PMID: 24906918).

11. More detailed annotations and panel titles within figures would assist readability

We have expanded descriptions in the relevant Figure legends (Figures 4 and 6) however we have not added additional information to the figures themselves because we feel that this would further clutter the already crowded figures.

12. The importance of the findings reported could be more readily appreciated with inclusion of some relevant mechanistic diagrams. For example the succinate significance in the TCA cycle could be emphasized. Further, the HIF-1a and the ROS-related actions could be depicted in diagram.

We thank the Reviewer for this useful suggestion. We have now added a new Figure 8 to our manuscript with the diagram summary of the key findings from our study.

Figure 8. Naked mole rats have distinctive cardiometabolic and genetic adaptations to their underground low-oxygen lifestyles- schematic summary of the key study findings.

Reviewer #3 (Remarks to the Author):

The paper by Faulkes et. al., sets out to investigate the evolutionary adaptations to anoxia and attempts to functionally characterise naked mole-rat's superior protection against ischaemia/reperfusion injury as compared to mouse. The authors first take a comparative approach and perform an extensive RNAseq analysis comparing naked mole-rat gene expression in the heart to 7 other species in the same genera as well as a divergent subterranean rodent and the mouse. The authors also do a more targeted comparative metabolomic analysis with the same species. This dataset is an extremely powerful and valuable resource, which not only supports the authors subsequent mechanistic investigation but more broadly provide highly useful dataset relevant to a wide range of researchers in the field of hypoxia research, cardiovascular disease research, naked mole-rat research, functional genomics research etc.

Considering that access to these animals and their tissues is not so simple as well as the table the authors provide to morphology and life history of these animals, I must stress how valuable this dataset is.

The subsequent experiments that were carried out by the authors to gain mechanistic understanding in how the naked mole-rats protect their hearts against ischaemia/reperfusion injury is however, weak in some parts, containing certain gaps, omissions and control flaws that I have pointed out in the comments below.

These need to be addressed to strengthen the conclusions made in the manuscript especially those pertaining to the idea that reduced succinate accumulation is responsible for reduced ROS and protection against injury, a mechanism that the authors seem to stress a lot and bring to the forefront in this manuscript.

In addition, this 'lowered succinate' phenomena has been described as reverse electron transport, however the authors, although elude to this process never overtly describe it. I am curious why RET is never mentioned as it can help put the mechanism brought forward by the authors into a greater context of processes already known about ischaemia/reperfusion in other mammals.

We thank the Reviewer for their comments. We have not measured ROS in NMR hearts in this study therefore we cannot comment directly on the possibility of RET driven ROS in NMRs. Whilst we already reference the work by Chouchani et al 2014 demonstrating the link between succinate accumulation and ROS in I/R (which included authors of the current work: Murphy, Aksentijevic), we take Reviewer's point on board and have now added a comment about RET on page 8.

In general also, I think the comprehensive comparative omics analysis provided by the authors could be made use of in a more resourceful way to the study's advantage and support some of the functional observations reported in the second part of the manuscript.

For example, why is the link to lactate efflux and elevated MCT transporters never explicitly explored by the authors? Or, is there something in the RNAseq that can explain the lack of succinate accumulation?

We agree with the Reviewer that we did not fully exploit omics data sets reported. We have now done so and we find that MCT1 expression is increased in NMR vs C57/BL6 mice (New Figure 7B) and that the gene for MCT2 is also upregulated in the RNAseq data for NMR compared to other mole rat genera (Figure 3). We have now added discussion on these points to our manuscript Results and Discussion sections. Specifically, in our Discussion (Page 8) we now include a reference to our previous study identifying the MCT1 transporter as a mechanism of succinate efflux post-ischemia (PMID: 32766828) and state that elevated MCT1 transporter protein expression in the NMR heart could allow for increased capacity to efflux succinate post ischemia.

Furthermore, we have also now analysed the cardiac effluent samples that were collected during the Langendorff experiments. This enabled us to now make direct comparison between NMRs and mice in their lactate efflux during normoxic perfusion, immediately after ischemia (1 min reperfusion) and 1 hour into reperfusion. These findings are now summarised in our new Figure 7A in the main manuscript (lactate efflux, MCT1 and LDH expression) and a new paragraph on page 6.

New Figure 7

We find that NMR hearts efflux less lactate at baseline in normoxia as well as at 1min and 1h post reperfusion, compared to C57/BL6 mouse (Figure 7A). However, during ischemia NMR hearts have elevated intracellular lactate compared to mice (Figure 6G) likely due to increased availability of carbohydrate reserves (glycogen, Figure 4C) which is broken down to yield increased endogenous glucose pool (Figure 6F) and increased endogenous lactate arising from glycolysis (Figure 6G). NMR hearts display increased MCT-1 expression (Figure 7B) and elevated myocardial LDH expression (Figure 7C), therefore the observation that NMR hearts efflux less lactate than C57/BL6 hearts during reperfusion (Figure 7A) is paradoxical. MCT-1 and LDH are also expressed in the mitochondria and therefore endogenous lactate could also be converted to pyruvate and used as a metabolic fuel rather than being effluxed as metabolic waste.

I think this study is remarkably valuable and provides very interesting insights to the naked mole-rat's extreme and superior physiology when it comes to dealing with low O₂. With a bit more effort to strengthen the claims, I think this would be a superb addition to our understanding of adapted physiology in hypoxia.

We thank the Reviewer for their comments. We hope that our additional experiments and manuscript amendments have improved our understanding of physiological adaptations to hypoxia.

Major comments:

1. (Line 145) Is there a summary about the parameters mentioned here like burrow and soil types, degrees of sociality, lifespan and hypoxia tolerance for each species investigated in this study. This could be added to table S1 and provide comprehensive set of parameters to complement the RNAseq for functionally linking phenotype and genotype via comparative approach.

We discussed this at length when preparing the manuscript and trying to find meaningful comparable data across species. For example, distance from mole hill to nest chamber is relevant but has never been measured in the wild. Other burrow measures are hard to relate in a meaningful way to O₂/CO₂ levels within the burrow as there are so many variables, therefore we cannot add these points to the Table. Sociality, group size and group biomass are already in Figure 1. Lifespan is presented in Table S1. Soil type has now been added to Table S1. Unfortunately, hypoxia tolerance has not been measured in all the species in the study in comparable ways.

2. (Line 180) The authors investigate the most highly expressed genes in NMR to be mitochondrial ETC genes. The authors bring this up again in the Discussion. How does this observation relate to previous reports that mitochondrial respiration from isolated heart mitochondria is lower in the naked mole-rat and the activity of the complexes is also lower (eg. the study from Lau et al., 2020 <https://doi.org/10.1016/j.cbpb.2019.110375> [19]).

This is an intriguing point. Of course, highly expressed mitochondrial genes do not necessarily translate to enhanced mitochondrial protein expression and respiration. Consistent with this, in Figure S3 we have shown that expression of mitochondrial complex subunits in NMR is comparable to C57/BL6. Furthermore, our cardiac function data shows significantly lower LVDP and HR in NMR at baseline vs C57/BL6 mice, which is consistent with the lower activity of NMR mitochondria reported by Lau et al. The relatively low basal cardiac function and mitochondrial respiration are ecophysiological adaptations to life in an energetically taxing environment and are well-described and discussed in NMR literature (PMID: [24363308](https://pubmed.ncbi.nlm.nih.gov/24363308/)). The reasons for the elevated RNA levels are unclear. One possibility is that the pool of RNA available might enable the rapid production of mitochondrial proteins without having to use energy to make RNA, in response to environmental energy demands. We also note that in comparison with the mouse, NMR cells have higher rates of autophagy (PMID: [24082110](https://pubmed.ncbi.nlm.nih.gov/24082110/)), perhaps giving NMRs the capacity to rapidly replenish damaged mitochondrial proteins. However, these possibilities are at this stage quite speculative.

3. Furthermore, I am not so sure about the implications of looking at the most abundant genes. Even if they are the most expressed genes, is the absolute expression levels between mouse and naked mole-rat different for these genes?

Thank you for this point. We already include absolute expression values in our data sets including the full RNA seq analysis provided in the supplementary Excel file (Supplementary data S2).

Is the protein also more increased for these genes? In order to make conclusions for very differing heart physiology based on top 30 expressed genes, I would be interested to know how most abundant genes between eg. human and mouse compare? Or perhaps more accessible for these authors, did the authors perform 30 most abundant gene analysis between the other African mole rats and NMR, or versus other African mole rats within themselves and against mouse.

The protein expression analyses were to examine the link to the set of highly expressed genes. The comparisons suggested by the reviewer with other mole-rats and the mouse are included in supplementary Data file S2 which shows the 30 most expressed genes (normalised as FPKM) across the sample sets. The Table cells are coloured to show shared expressed top 30 genes compared to the NMR. Unique NMR genes are in yellow and a brief description was given in the text “Comparison of the top 30 expressed cardiac genes across all the species analysed exemplifies the unique gene expression pattern of NMRs. Only 13 genes listed in the NMR top 30 are found in the top 30 of the other species, and the top 10 in the NMR are not shared across the other species (Supplementary Information Data S1)”. We have edited manuscript further to now read: “**Comparison of the top 30 expressed cardiac genes across all the species analysed exemplifies the unique gene expression pattern of NMRs and their divergent phenotype when compared to other mole-rats, and also to the mouse and golden mole outgroups. The top 10 expressed genes in the NMR do not appear within the top 30 from any other species while many of the top 10 genes from the other species are shared in common (Supplementary Information Data S2).**” (page 5)

4. (Line 220) The authors measure plasma insulin levels but do not include how they measured insulin in Materials and Methods. Naked mole-rat insulin has mutation effecting the beta-sheet and is notoriously resistant to measurement by mouse and human ELISAs. If the authors used mouse or human kits, the much lower insulin level could be due to detection efficiency as ELISAs are antibody based and therefore would not detect the naked mole-rat version of insulin. Please check, elaborate and validate that the insulin is really detected with similar efficiency.

NMRs have been previously shown to have unique pancreatic physiology, low blood glucose levels (as also shown in our study), low or undetectable insulin (as also shown in our study) and an impaired glucose tolerance test, maintaining elevated glucose levels for more than 2 hours when challenged with a supraphysiological dose. If given a dose of human insulin of physiologically relevant dose, blood glucose rapidly drops to extremely low levels and remains there for several hours (PMCID: PMC7149588). The exquisite sensitivity to human insulin coupled with the abnormal glucose tolerance test and low levels of insulin suggest that NMRs naturally only produce small quantities of this hormone and that this is usually sufficient for their needs (PMCID: PMC7149588, PMID:15560867). We however agree with the Reviewer that the insulin kit used by MRC Biochemistry Laboratory (Addenbrookes Hospital, Cambridge, MesoScale Discovery Rockville, MD, USA, K152BZC-3, details of the assay now included in our Methods) could potentially affect the detection of NMR insulin however our results are in line with the known NMR insulin insensitivity and impaired GTT response phenomenon.

5. Furthermore, I did not find in the materials and methods a description for how any of the 4 plasma metabolites were measured. Please include details of how glucose, insulin, triglycerides and lactate dehydrogenase were measured.

We apologise for not including full details of our plasma analysis that was performed by MRC Mouse Biochem laboratory at Addenbrookes Cambridge. Detailed description of methods used for the plasma analysis has now been added to the Methods section of the manuscript.

6. (Line 232) Although showing slightly differing results, there has already been a publication on HIF1alpha protein as well as mRNA expression in the naked mole-rat compared to mouse in the heart. The strain of mouse used was different perhaps explaining the difference in observations. The authors should at least cite this paper and omit the word unexpected since stabilised HIF1alpha protein expression has already been shown in the study by Xiao et al, 2017. <https://doi.org/10.18632/oncotarget.22767> [20]

We thank the reviewer for this suggestion. This paper has now been cited and the word “unexpected” removed.

7. (Line 241) The LDH1 and MCT1 blots shown in Suppl Fig 1 showing the expression of Hif1alpha targets are consequently overexpressed has a few issues:

a. Although stated that MCT1 is overexpressed the representative blot and the band density quantification (Suppl Fig 1D) shows non-significance.

Reviewer was correct, and the P value is 0.048. This has now been amended and included in new Figure 7.

b. Why do the authors use different housekeeping proteins in every blot that they do? In any case it is questionable whether the use of housekeeping/loading controls makes sense in comparing across species or rather adds a confounding factor since it seems like none of the housekeeping proteins chosen actin (Figure S1a), Vinculin (S1c) and Tubulin (S1E) have consistent expression between mouse and naked mole-rat. They all seem to have species specific differential protein expression and therefore are not appropriate loading controls. If the proteins were loaded at the same concentrations, perhaps foregoing the normalisation based on housekeeping protein and show equal loading based on a whole protein staining like Ponceau is more appropriate for across species comparison.

We thank the Reviewer for this constructive comment. We agree with them and we have also encountered similar issues previously. However, there are numerous studies that use and demonstrate their results using commonly shared housekeeping genes such as Actin, Tubulin or GAPDH simultaneously in both mice and NMR (e.g., PMID: 34351155; <https://doi.org/10.1021/acs.jproteome.1c00131>; <https://doi.org/10.1371/journal.pone.0035890>; <https://doi.org/10.1074/mcp.RA120.002301>; <https://doi.org/10.1016/j.stemcr.2017.10.001>; <https://doi.org/10.1073/pnas.0809620106>; <https://doi.org/10.1038/s41586-020-2410-x>).

Supporting this, inserted below are representative western blots (HIF1a expression) between NMR and C57/BL6 mouse samples by immunoprobng of the housekeeping gene Tubulin.

In order to compare HIF1alpha protein levels between NMR and C57/BL6 mice we performed the following normalisation procedure. Protein samples were resolved (4-20% PAGE), electroblotted and stained for total protein with Ponceau S. Initial densitometry analysis of the Ponceau S signal per lane for each initial sample enabled the samples to be diluted to generate equal intensity Ponceau s signal. These samples were then resolved by PAGE (4-20%) electroblotted and exposed to the same antibodies (two different anti-HIF1a antibodies and anti-Tubulin antibody) used previously in this study. Thus, using the Ponceau S staining procedure mentioned by the Reviewer shows that our use of Tubulin as a loading control is valid in this instance.

c. The most direct consequence of HIF1alpha stabilisation would be the RNA expression of HIF1alpha targets rather than protein expression. In order to make the point that HIF1alpha targets are upregulated in the naked mole-rat, mRNA for target genes should be quantified for known targets like VEGF, GLUT1, LDH and MCT1. This could already be ascertained from the RNAseq data (mouse versus NMR) the authors performed but did not include in the supplementary data set.

To address the Reviewer's question related to upregulation of HIF-1alpha specific genes in NMR and C57/BL6 mice, a data summary of representative genes is now provided in a Table below. We looked specifically in the DEG expression data detected in our analysis for HIF-1 target genes encoding metabolic enzymes (**blue font**), and transporters (**red font**) involved in glucose metabolism. Expression (up/down) with respect to the NMR of the various species, NS = gene not significantly up or down.

Whilst there are some changes in DEG of genes downstream of Hif-1alpha, RNAseq only provides a snapshot of the gene expression in the tissues collected in normoxia. Thus, we will not see all the downstream target genes being affected as would be the case during hypoxia.

Some of this data (HIF1 alpha target genes) is already presented in our volcano plot (Figure 3). All the RNA sequencing data is also available in Supplementary Data Set 2 as well as is deposited on Dryad (open access).

Gene	Encoded protein	NMR vs Damaraland Mole-rat	NMR vs Cape dune Mole-rat	NMR vs Cape Mole-rat	NMR vs Common Mole-rat	NMR vs Highveld Mole-rat	NMR vs Mahali Mole-rat	NMR vs Natal Mole-rat	NMR vs Golden mole-rat
ENO1	Enolase 1	down	NS	NS	NS	down	NS	NS	NS
ENO2	Enolase 2	NS	up	up	NS	down	down	down	down
LDHA	Lactate dehydrogenase A	down	down	down	down	down	down	down	down
PFKFB2	6-Phosphofructo-2-kinase/fructose-2,6-biphosphatase 2 (cardiac type)	up	up	up	up	up	up	up	up
PFKFB4	6-Phosphofructo-2-kinase/fructose-2,6-biphosphatase 4 (testes type)	NS	NS	up	up	up	up	up	up
SLC2A4	Glucose transporter 4	NS	NS	NS	NS	up	NA	up	up
VEGFA	Vascular endothelial growth factor A	NS	NS	NS	NS	NS	NS	NS	NS
VEGFB	Vascular endothelial growth factor B	NS	NS	down	NS	NS	NS	NS	NS
VEGFC	Vascular endothelial growth factor C	NS	down	down	down	down	down	down	down
VEGFD	Vascular endothelial growth factor D	down	down	NS	down	NS	NS	down	down
SLC16A4	Monocarboxylate transporter 5	NS	NS	up	up	NS	NS	NS	up
SLC16A7	Monocarboxylate transporter 5	up	up	up	up	up	up	up	up

8. (Line 255) The authors use the word normothermic ischaemia. Please define the exact temperatures used as the general reader is not aware of normothermic values for naked mole-rats, especially as their thermoneutral zone is a wide range of 28-32 degrees.

We apologise for not making this clearer. It means 37C (used for NMR and C57/BL6) and these conditions are the same ones used in previous I/R studies (Chouchani et al 2014 Nature, Prag et al 2021, Cardiovascular Research, Prag et al Circulation Research 2022). We have now clarified this in the methods.

9. The influence of temperature may impact the extent of injury of the heart. Therefore the mouse undergoing normothermic ischaemia at 37 degrees could sustain greater damage purely due to elevated temperatures compared to the naked mole-rat undergoing normothermic ischaemia at 32 degrees or less. It is well known that reducing temperature during ischaemia protects the tissue at risk. I understand naked mole-rats may be precious and limiting, so to control for the significant confounding factor, the authors should perform the ischaemia/reperfusion protocol on mouse hearts under the same temperature conditions used for the naked mole-rat and assess the injury thereafter. Likewise, the metabolites measured during normothermic ischaemia may be a result of slower metabolism due purely to lower temperatures. Metabolites, like succinate, other TCA cycle intermediates and lactate should also be measured in mouse ischemic hearts under temperature conditions used for the naked mole-rats.

All experiments were performed at 37C (identical perfusion conditions to Chouchani et al 2014 Nature, Prag et al 2021, Cardiovascular Research, Prag et al Circulation Research 2022) and we apologise for not making this clearer. Temperature has now been added to the manuscript (Methods).

10. Figure 4C is shows a representative slice from the heart. Since the naked mole-rat hearts are different sizes and the injury zone can be located in slightly different sites, can the authors provide images of the entire heart, i.e. all 4-5 slices and elaborate whether injury size in figure 6D was quantified from a distinct slice or the entire heart?

Multiple heart slices were used for quantification of the infarct sizes; however for the figure we used a representative slice from one of the NMR hearts. NMR hearts are smaller and more fragile than C57/BL6 mouse hearts thus were challenging to slice and fewer slices were obtained using the Zivic slicing gauge (as stated in our methods and used for our previous work PMID: 35959683, PMID: 32766828).

C57/BL6 Representative sections from 1 heart

NMR sections from 3 hearts

Representative image in new Figure 6D

11. (Line 255) The authors state that the NMRs have enhanced energy supply by showing total adenine pool. Why are the authors only reporting total adenine pools? From Figure 4a, it seems like the authors are able to measure at least separate ATP and AMP pools. The ratio of ATP to ADP or AMP would be a more useful and insightful measure of whether the naked mole-rats actually maintain higher energy supplies. If the ¹H-NMR analysis performed allows separation of adenine pools, please reanalyse and show these separately.

Unfortunately, we are not able to distinguish ATP and ADP by nuclear magnetic resonance. The AMP data is deposited in the publicly available dryad data depository and shows no difference between NMR and C57/BL6 mouse.

Faulkes, Chris G. et al. (2023), Naked mole rats have distinctive cardiometabolic and genetic adaptations to their underground low-oxygen lifestyles,
Datasets: <https://doi.org/10.5061/dryad.w9ghx3fts>; <https://doi.org/10.5061/dryad.66t1g1k66>

12. (Line 257) The authors report reduced succinate accumulation, however they do not provide a baseline measurement of succinate levels of ex vivo perfused hearts prior to ischaemia. Although there is succinate measurement from freshly dissected hearts from mouse and naked mole-rat in Figure 4f which can hint at baseline levels, we cannot conclude that the same amounts exist in isolated perfused hearts at the end of 20 minute equilibration period. The authors should provide metabolite measurements for naked mole-rats and mice either after the equilibration period just prior to ischaemia exposure or perhaps as a better control after 40 minutes of normoxia perfusion (20 minutes equilibration and 20 minutes normoxia versus 20 minutes ischaemia). Furthermore, the units for Succinate in figure 4F and Fig 6H are in different Units (normalised peak area versus arbitrary units). Can the y-axis be made consistent between different figures for the metabolites.

We have provided extensive succinate measurements for C57/BL6 mouse hearts in many other publications (PMID: 35959683, PMID: 32766828) as well as under normoxia in Figure 4. Regarding units, we apologise for this lapse. All plots have now been normalised in the same way (to peak area) and we have relabelled axes in Figure 6 so they are all comparable.

13. Alternatively, if access to naked mole-rats may be a problem, but optimally in addition to the experiment in point 12, evidence of succinate accumulation in ischaemia should be further supported by looking at other TCA cycle intermediates. In another publications by some of the same authors of the current article (Couchani 2014), citrate, aconitate, alpha-ketoglutarate, fumarate and malate have been shown to be unchanged under ischaemia relative to normoxic conditions whereas succinate accumulates 12-fold. Therefore, the authors should put in effort to show succinate in relation to the other TCA cycle intermediates and their changes relative to normoxic conditions to really pinpoint the lack of succinate accumulation and not just a reduced overall mitochondrial metabolism in the naked mole-rat.

14. The authors claim that it's succinate accumulation which leads to ROS and IR injury and that by having reduced succinate accumulation under ischaemia, the naked mole-rats are therefore protected against injury. ROS however has not been measured in the naked mole-rat under IR conditions in order to make the link. Therefore, the conclusion the authors make here based on weakly controlled succinate accumulation experiment and no ROS measurements is unsubstantial this point. To strengthen this conclusion, the authors should measure ROS in baseline and ischaemic/reperfused tissue and compare accumulation of ROS between naked mole-rat and mouse.

As discussed in an earlier comment regarding RET, the aim of our study was not to measure succinate metabolism in relation to RET and ROS. The link between succinate concentration and ROS production in IR injury is well -supported in mouse hearts, but we agree with the Reviewer that the ROS-succinate relationship in NMR hearts has not been demonstrated. Therefore, we have adjusted the text on page 8 to indicate that this is an interesting possible mechanism to explore in future studies of NMR.

15. The authors claim that naked mole-rats accumulate lactate which then gets rapidly effluxed in the first minute of reperfusion. The authors don't go so far as to overtly say this may additionally be a protective mechanism, but by showing this piece of data imply it. However, the presented data is incomplete in its current state to make any reasonable conclusion about whether naked mole-rat have greater efflux of lactate and therefore lactate's role in IR injury. The authors can strengthen this very important observation with the following experiments.

a. The authors first claim that naked mole-rats accumulate lactate during ischaemic phase (Fig6G). The authors however only show lactate levels in ischaemic phase and it therefore cannot be concluded with certainty that the lactate has indeed accumulated compared to baseline levels. This can be amended by the experiment I suggested in Point 12.

b. The authors show that at baseline, the lactate in the effluent is the same in mouse and naked mole-rat (Suppl. Fig 6a). However, the measurements for lactate after reperfusion (Suppl. Fig 6b) is only shown for the naked mole-rat. Why don't the authors show lactate in the effluent for the mouse as well? Without comparing lactate in effluent after reperfusion to the mouse, we cannot draw any conclusion from the dataset. A such, we do not know whether a species, like mouse, which is not protected from ischaemia, also effluxes lactate at reperfusion and restores normal levels quickly or whether this is indeed what differentiates the naked mole-rat from mouse and is therefore unique to the naked mole-rat hinting that it may be contributing to the resistance to IR injury in this species. Generally, how naked mole-rats deal with lactate accumulation during ischaemia or at reperfusion thus avoiding the problem of lactic acidosis is a highly fascinating topic and the authors here can uncover a potentially interesting phenomena that can contribute to our understanding of protective heart physiology under ischaemia. Therefore, providing evidence that the naked mole-rat has a greater capacity for efflux could be extremely strengthening for this manuscript. If the dataset is completed as I suggest above, this figure can come out of supplementary and into the main manuscript.

We have followed the Reviewer's suggestion and performed additional analyses of effluents from the Langendorff experiments which are now shown in a new Figure 7 (lactate efflux, MCT1 and LDH expression). We can now make a direct comparison between lactate production and efflux in normoxic perfusion, immediately after ischemia (1 min reperfusion) and 1 hour into reperfusion (NMR vs C57/BL6).

We find that NMR hearts efflux less lactate at baseline in normoxia as well as at 1 min and 1h post reperfusion, compared to C57/BL6 mouse (Figure 7A). However, during ischemia NMR hearts have elevated intracellular lactate compared to mice (Figure 6G) likely due to increased availability of carbohydrate reserves (glycogen, Figure 4C) which is broken down to yield increased endogenous glucose pool (Figure 6F) and increased endogenous lactate arising from glycolysis (Figure 6G). NMR hearts display increased MCT-1 expression (Figure 7B) and elevated myocardial LDH expression (Figure 7C), therefore the observation that NMR hearts efflux less lactate than C57/BL6 hearts during reperfusion (Figure 7A) is paradoxical. MCT-1 and LDH are also expressed in the mitochondria and therefore endogenous lactate could also be converted to pyruvate and used as a metabolic fuel rather than being effluxed as metabolic waste.

We have amended our manuscript to include new findings and conclusions (Results, Discussion). We have also incorporated Reviewers remarks into our manuscript (Discussion).

“At present it is unknown whether a species, like mouse, which is not protected from ischaemia effluxes lactate at the same rate as NMR. Our finding of lower lactate efflux in NMR heart may suggest improved use of lactate as fuel and may be contributing to the resistance to I/R injury in this species. How naked mole-rats deal with lactate accumulation during I/R thus avoiding the problem of lactic acidosis is therefore a highly fascinating phenomenon which our study potentially uncovered thus contributing to our understanding of protective heart physiology under ischaemia.” (page 9)

16. (Line 405) In methods section, please provide details to the temperature used for Langendorff experiments for each species

This has now been added (37C).

17. Supplementary Dataset 2 provides complete dataset for all genes for NMR versus all other African mole rat species. Why is the dataset for naked mole-rat versus mouse not provided especially since the rest of the manuscript focuses on mouse against naked mole-rat comparison? I think including this dataset would complement the rest of the study.

We also agree - unfortunately it was not possible to run a full DEG analysis of the NMR versus the mouse. We spent time trying to do this but the analysis pipeline would not run due to annotation problems from mapping the mouse to the NMR genome and *vice versa*. We outsourced this to the BGI Genomics bioinformatics team (company who performed RNA sequencing) and they were also unable to perform the analysis. The closest that we have been able to generate is a full list of normalised values for gene expression in NMRs and the mouse (the top 30 of which are displayed in Figure 2C and D). This is now included as Supplementary Data S1 (with original Data S1 becoming S2 and S2 becoming S3).

Minor comments

1. Line 142 Mahali mole rat needs a hyphen for consistency

This has been changed. Thank you for your suggestion.

2. The Authors claim in the legend for Fig 6 that n=5 is used for ascertaining the infarct size, when in fact from Figure 6D only 2 points are visible for naked mole-rat and 3 points are visible for the mouse.

Apologies that data points in stacked bar charts fall on top of each other- we have now re-plotted to show all individual points (New Figure 6D).

3. Paragraph beginning with Line 307 should cite the Xiao et al, 2017 paper as well.

This paper has now been referenced in our manuscript. Thank you for your suggestion.

4. (Line 352) claims that the study provides insights into the NMR cardiac adaptations to the dark and O₂. It is clear that the manuscript handles adaptation to low oxygen but the significance of this finding to the dark is less clear, especially as the holding conditions for NMRs described has a light/dark cycle. Please omit or elaborate.

We have removed the reference to the dark in line 352 as we have not studied this.

Reviewer #4 (Remarks to the Author):

In this study RNAseq and metabolomics have been performed on the cardiac tissues of naked mole-rats and other mole-rat genera, providing fascinating genetic and metabolomic insight into the unique characteristics of the naked mole rat. I do however have a number of comments to be addressed, primarily surrounding the metabolomics methodology.

1. A bit more detail is needed on the extraction, storage and processing of the heart samples to demonstrate sample integrity was maintained for downstream analysis. Some sections of the method indicate naked mole rat and other mole rat samples were not subjected to the same analysis (e.g. the cardiac perfusion). It is not clear exactly what processes the samples went through prior to analysis and whether they were in fact treated exactly the same, so please clarify this. Any variation in sample treatment could have significant effects on metabolites, creating artificial differences.

We thank the Reviewer for this point. We have now included additional sections in our methods to clarify how samples were collected from the wild-caught animals. In summary, all samples were collected in RNA later tissue preservation buffer, stored at -80C and handled in liquid nitrogen during preparation. All C57/BL6 mouse samples were collected using the same tissue collection approach (RNA later followed by liquid nitrogen freezing) to facilitate comparison with NMR samples.

2. The authors state that NMR analysis was “semi-targeted”. Can the authors expand on this to describe how they selected the metabolites discussed? Were specific analytes the focus of this study from the start or was some kind of statistical framework used to select analytes to identify and compare?

We thank the Reviewer for this point. Specific analytes were not selected for the statistical analysis. We used all metabolites that could be confidently assigned by nuclear magnetic resonance. We used the term semi-targeted since metabolite coverage by nuclear magnetic resonance is much lower than other techniques such as LC-mass spectrometry. However, we agree with the reviewer that we did not select specific metabolites and we have therefore re-phrased this as untargeted.

3. Were there any statistical differences in other metabolites that were either not discussed or not identified? Although not discussed, this could be useful and interesting information to share.

All identified metabolites are presented in the heat map and were used for PCA analysis, given in the online data dryad depository.

Faulkes, Chris G. et al. (2023), Naked mole rats have distinctive cardiometabolic and genetic adaptations to their underground low-oxygen lifestyles,

Datasets: <https://doi.org/10.5061/dryad.w9ghx3fts>; <https://doi.org/10.5061/dryad.66t1g1k66>

4. How many metabolites were used in the LDA and PCA plots - all detected features or just the metabolites listed on the heatmap in Fig 4A? Please specify in the legend or methods.

All detected metabolites were included in the LDA and PCA plots. We have now clarified this point in the legend.

5. Regarding the LDA plot in figure 4B, presumably the ellipses depict a confidence interval. If so, this should be detailed in the figure caption.

Yes, these ellipses depict 95% confidence intervals for the LDA analysis and 90% confidence intervals for the PCA analysis in the supplemental material. We have now added these details to the figure captions.

REVIEWER COMMENTS

Reviewer #2 (Remarks to the Author):

REVIEWER 2 (Summary of Author responses (commentary and/or edits))

Responses to Items 3,4,5,7, 9 & 11 are adequate.

Item 10 becomes redundant.

Item 12 new figure provided (Fig 8) has been well executed.

Item 2.

The original comment: The use of terminology 'eusocial' is not very helpful (unusual term). A better description would be more specific, highlighting the animal COLONY SOCIAL organizational features which confer extreme phenotypic RESPONSIVENESS.

The point of the comment was not to challenge the definition of the NMR as eusocial – simply to request that a short phrase of description be added to assist reader (in the Abstract and the introduction. This would be of value to a broad range of non-expert readers. For instance, based on the Author responses a short bracketed comment noting that “... the NMR is a eusocial mammal (exhibiting colony hierarchical structure and highly compact habitat conditions) ...” would be sufficient.

Item 8. The additional information is useful. Perhaps even some speculation allowed at Editor's discretion – given the ETC component transcripts were upregulated but translation not, some conclusion about longevity in hypoxia would be of interest.

REVIEWER 1 (Comments regarding responses to Reviewer 1 as requested by the Editor).

Responses to Remarks / Items 1, 3, 4, 5, 6, 7, 8, 9, 10 are adequate.

Item 2. The reviewer question could be usefully addressed by a brief general comment relating to the limitations on drawing individual comparisons between various mole species, even if the specific comparison suggested by the reviewer is not particularly relevant.

Item 11 This is a particularly strong response and the new data are an important addition to the paper.

Minor points: 1 -10 all addressed adequately.

Reviewer #3 (Remarks to the Author):

Following the revisions, this manuscript's strength continues to be the valuable comparative RNAseq dataset across all African mole-rats. I have to stress that a deeper analysis of the similarities and differences through for example enrichment analysis, GSEA etc , could shed some light and reveal novel insights into the adapted heart physiology in the naked mole-rat. Whether or not the authors do this or provide the data for others in the field, I believe the

collection of samples and analysis provided thus far was a big undertaking and resulted in a highly valuable resource.

The second part of the paper which attempts to link some of the molecular findings to function continues to, in some instances, lapse in proper controls, this especially goes for the succinate measurement which is one of the central claims of the paper as it is also included in the final summary diagram in Figure 8. Some other findings continue to be highly speculative without any supported data. For example, increased succinate flux, which is also featured in the schematic summary in Figure 8, is not at all experimentally addressed. Other data, in my opinion, was not properly interpreted such as the MCT1 protein expression which on the western blot in Supplementary figure S1. C looks the same or even lower in naked mole-rats. The authors base their quantification of MCT1 on normalisation to Vinculin (which does not have stable expression between mouse and naked mole-rat), and therefore report MCT1 to be more highly expressed. This result is also part of the overall summary highlighting key findings, however the evidence for high MCT1 expression is questionable.

As such, although all of the findings presented in the manuscript are potentially of high interest, the individual components are supported by weak evidence. I therefore have reservations about the findings and the final general model put forward in Figure 8. The details to my concerns to individual points are below.

1. To the added comment on RET and ROS, why have the authors not measured ROS in their study?

2. To the comment about mining RNAseq data, MCT1 expression and succinate efflux:

Why has succinate efflux not been tested? As the authors have collected the effluent and have the means to measure succinate, the authors should analyse whether indeed succinate is more readily effluxed supporting their speculation and their final schematic diagram (Fig 8).

MCT1 expression was analysed by western blotting. The blot in supplementary figure 1C looks like MCT1 is either the same or even lower in the naked mole-rat. The authors use Vinculin as housekeeping protein, but this protein from the blot shown seems to have a species specific expression as well, where it looks like vinculin is less expressed in naked mole-rat hearts. This issue was raised by me in the first round of revision and the authors have replied below that Tubulin seems to be a better housekeeping protein so perhaps Tubulin should be used to normalise MCT1. I understand Tubulin size is similar to MCT1, however the authors could strip and reprobe their MCT1 blot or run a parallel gel with the same samples. Caution has to be exercised when normalising to a housekeeping gene/protein to compare across species and choose a protein that is truly expressed equally in the two species. Finding such a protein is challenging. If such a protein is not found, it is best to avoid normalising and rather rely on whole protein staining via Ponceau or Coomassie. Assuming the authors attempt to load equal protein concentrations for their samples, and judging from the MCT1 gel alone, I cannot conclude a 3 fold higher MCT1 expression as quantified in figure 7B.

To conform this finding, the authors could perform a qPCR and a western blot with whole protein staining as control.

3. To the new Figure 7

The sentence about MCT1 and LDH in mitochondria is highly speculative. This idea is still highly controversial and the authors have not provided a reference for this statement. The authors should back this up with a quantification of MCT1 and LDH from mitochondrial fractions.

In address of Major comments:

Point 4:

Unless more precise validation is done for detection of insulin with the mouse/rat kit, for example blood insulin after overnight fast versus refeeding versus bolus of glucose, the results from a mouse/rat based insulin kit may give false negatives. Alternatively the authors can use a guinea pig based ELISA kit as the naked mole-rat insulin has a similar mutation as all other hystricognaths including guinea pigs.

Point 7c: In interpreting the table with Hif1 targets in different mole-rats. Is HIF1 alpha also stably expressed in other mole-rat species? This information would help to make sense of the Table supplied.

Point 8:

Does running the experiment outside of naked mole-rat thermoneutral zone, ie. at temperatures much higher 37 rather than 32 affect heart function?

Point 10:

To support the minor damage post-ischaemia in the naked mole-rat heart which is a remarkable finding, do the authors have LVDP measurement for the reperfusion interval. What was the % recovery from the baseline measurement reported in Suppl Figure S5.

Further comment on Point 10:

I am surprised by how small the naked mole-rats hearts are compared to body size. In the paper by Grimes et al, (PMID: 24363308) reports a heart:body mass % 0.41 for naked mole-rat and 0.5 for mouse. The above publication is therefore in discrepancy of the results presented here. A 3 fold smaller heart proportionally is noteworthy and would have a significant impact on heart function. Can the authors comment on this. Furthermore in the Table S1 where the authors provide details for all African mole-rats, all the hearts from the mole-rat species seem strangely small. Is this a feature of the African mole-rat family? I was particularly surprised by the Suillus mole-rat where the authors report a heart weight of 88.1mg for a 639g animal compared to for example Cryptomys h. h. with an 82.6 mg heart and weighing just 84 g. Could the authors check this data please and if this is really the case, this is quite astounding physiology given that generally heart:body weight is proportional across mammals (see figure 32.1 from <https://doi.org/10.1016/B978-0-12-385471-1.00032-5>)

Point 11:

In Figure 4a, in the heatmap, ATP comes up as a separate metabolite. I am assuming for the ischaemic hearts, the authors used a similar method. Why can't the adenine pools be

separated now?

Furthermore thank you for pointing to the datasets, unfortunately the two links don't work and bring up a page saying DOI not found. Please check the link.

Point 12:

Here I was more interested in the naked mole-rat heart and not the mouse heart which indeed has been reported several times. I am still not convinced whether the naked mole-rats do or do not accumulate succinate with ischaemia. The references provided by the authors emphasize the importance of looking at the same tissue set-up for succinate measurements, where one study looked within the same heart slice at injured versus healthy zone PMID: 35959683 or performed what I suggested above, succinate measurement after equilibration before ischaemia, and during/directly after ischaemia PMID: 32766828. In order to state that there is no succinate accumulation we need a baseline measurement for naked mole-rats, considering especially that the baseline was analysed from hearts dissected directly from animals connected to their circulation and at 28-32 degrees (thermoneutral zone of naked mole-rat) versus succinate measurement from ischaemic hearts maintained ex vivo and at 37 degrees (a stressful temperature for naked mole-rat physiology). These are two very different set ups to make direct comparisons regarding succinate levels. Furthermore proportionately the succinate levels between mouse and naked mole-rat in baseline and ischaemia do not vary much.

Point 13:

This important point was not addressed by the authors.

Point 14:

From the data presented the only conclusion that can be reached is that there is generally less succinate in the naked mole-rat, in both normal and ischaemic conditions. This may be because of lower/less efficient mitochondria, lowered heart rate/metabolic rate, or some other parameters. I am not sure how the succinate measurement in their set-up relates to ischaemia as we do not have a controlled baseline succinate measurement to compare succinate under ischaemia with. Please see my comment for Point 12.

Point 15:

MCT1 and LDH at mitochondria is highly speculative and should be supported by blots from mitochondrial fractions or omitted.

Point 17:

Such analyses have been performed previously for other tissues like liver and could help to support some of the authors findings for the rest of the manuscript which focused mainly on comparing mouse and naked mole-rat. Have the authors tried to consult previous literature of cross-species RNAseq DEGs analysis and tried to apply similar methods?

Such analyses have been performed previously for other tissues like liver and could help to support some of the authors findings for the rest of the manuscript which focussed mainly on

comparing mouse and naked mole-rat. Have the authors tried to consult previous literature of cross-species RNAseq DEGs analysis and tried to apply similar methods?

See DOI:10.1126/science.aau0236, doi: 10.1186/s12915-018-0546-z

Reviewer #4 (Remarks to the Author):

The authors have addressed all of my comments appropriately and I have no further changes to suggest.

REVIEWERS RESPONSE

Reviewer #2 (Remarks to the Author):

Item 2. The original comment: The use of terminology ‘eusocial’ is not very helpful (unusual term). A better description would be more specific, highlighting the animal COLONY SOCIAL organizational features which confer extreme phenotypic RESPONSIVENESS.

The point of the comment was not to challenge the definition of the NMR as eusocial – simply to request that a short phrase of description be added to assist reader (in the Abstract and the introduction. This would be of value to a broad range of non-expert readers. For instance, based on the Author responses a short bracketed comment noting that “.... the NMR is a eusocial mammal (exhibiting colony hierarchical structure and highly compact habitat conditions) ...” would be sufficient.

We thank the Reviewer for their clarification. We have now amended our manuscript to add the following phrases (in green).

Abstract

“The naked mole-rat *Heterocephalus glaber* is a eusocial mammal (with a social insect-like colony structure) with extreme longevity (37-year lifespan), extraordinary resistance to hypoxia and absence of cardiovascular disease.”

And in the **Introduction**:

“Naked mole-rats (*Heterocephalus glaber*, NMRs) are characterised by their social insect-like (eusocial) behaviour defined by a reproductive division of labour, overlapping generations and cooperative care of the young. They exhibit a suite of adaptations to living in a hypoxic (low oxygen, O₂) subterranean environment¹.”

Item 8. The additional information is useful. Perhaps even some speculation allowed at Editor’s discretion – given the ETC component transcripts were upregulated but translation not, some conclusion about longevity in hypoxia would be of interest.

We have now added this comment to our Discussion, page 9.

“During ageing in C57/BL6 mice (2.5 years), mild depolarization disappears in most tissues including the heart, however, age-dependent decreases in the levels of bound kinases is not observed in NMRs³². As a result, protein damage, which is substantial during the ageing of short-lived mice, is stabilised at low levels during the ageing of long-lived NMRs. It has been suggested that mild mitochondrial depolarization is a crucial component of the anti-ageing system³² contributing to NMR longevity. We also observed that highly expressed mitochondrial genes in NMRs do not necessarily translate to enhanced mitochondrial protein expression and respiration. The relatively low basal cardiac function and mitochondrial respiration could be part of ecophysiological adaptations to life in an energetically taxing environment and may also contribute to NMR longevity.”

REVIEWER 1 (Comments regarding responses to Reviewer 1 as requested by the Editor).

Responses to Remarks / Items 1, 3, 4, 5, 6, 7, 8, 9, 10 are adequate.

No response required.

Item 2. The reviewer question could be usefully addressed by a brief general comment relating to the limitations on drawing individual comparisons between various mole species, even if the specific comparison suggested by the reviewer is not particularly relevant.

We previously pointed out in our last rebuttal that the reviewer has misunderstood the NMR versus the golden mole comparison in the volcano plot. While individual comparisons between two species might be argued as limiting, we have done a comparative analysis across the phylogenetic tree which shows that the NMR is different from all the other species in our data set and mitigates against the limitation of single species comparisons. Moreover, for differential gene expression analysis, fold changes in expression can only be calculated between pairs of samples or species.

Item 11 This is a particularly strong response and the new data are an important addition to the paper.

We would like to thank the Reviewer for finding our response strong and new data important addition to the paper.

Minor points: 1 -10 all addressed adequately.

No response required.

Reviewer #3 (Remarks to the Author):

Following the revisions, this manuscript's strength continues to be the valuable comparative RNAseq dataset across all African mole-rats. I have to stress that a deeper analysis of the similarities and differences through for example enrichment analysis, GSEA etc, could shed some light and reveal novel insights into the adapted heart physiology in the naked mole-rat. Whether or not the authors do this or provide the data for others in the field, I believe the collection of samples and analysis provided thus far was a big undertaking and resulted in a highly valuable resource.

We thank the Reviewer for their comments and finding our data set valuable. Our data is deposited for future public access and thus we hope can be a useful resource for others to use in the future, especially as it contains hard to access non-model mole rat families.

The second part of the paper which attempts to link some of the molecular findings to function continues to, in some instances, lapse in proper controls, this especially goes for the succinate measurement which is one of the central claims of the paper as it is also included in the final summary diagram in Figure 8. Some other findings continue to be highly speculative without any supported data. For example, increased succinate flux, which is also featured in the schematic summary in Figure 8, is not at all experimentally addressed. Other data, in my opinion, was not properly interpreted such as the MCT1 protein expression which on the western blot in Supplementary figure S1. C looks the same or even lower in naked mole-rats. The authors base their quantification of MCT1 on normalisation to Vinculin (which does not have stable expression between mouse and naked mole-rat), and therefore report MCT1 to be more highly expressed. This result is also part of the overall summary highlighting key findings, however the evidence for high MCT1 expression is questionable.

As such, although all of the findings presented in the manuscript are potentially of high interest, the individual components are supported by weak evidence. I therefore have reservations about the findings and the final general model put forward in Figure 8. The details to my concerns to individual points are below.

We have amended Figure 8 (now Figure 7) and removed any reference to MCT and LDH.

New Figure 7. Naked mole rats have distinctive cardiometabolic and genetic adaptations to their underground low-oxygen lifestyles- schematic summary

1. To the added comment on RET and ROS, why have the authors not measured ROS in their study?

This unfortunately cannot be done as we don't have the technical ability to measure ROS in situ in the Langendorff perfused heart and we don't have access to tissue for additional biochemical analysis.

2. To the comment about mining RNAseq data, MCT1 expression and succinate efflux:

Why has succinate efflux not been tested? As the authors have collected the effluent and have the means to measure succinate, the authors should analyse whether indeed succinate is more readily effluxed supporting their speculation and their final schematic diagram (Fig 8).

We have performed additional analysis of the effluent which showed that NMR hearts efflux less succinate, in agreement with intracellular succinate also being lower. We thank the Reviewer for their suggestion as we believe this additional data has strengthened our manuscript further. We have now updated our manuscript [Figure 6J, methods, results and discussion (green font)].

Figure 6. Naked mole-rat heart is resistant to ischemic damage. (A) Representative naked mole-rat and C57/BL6 mouse used in the study. (B) Langendorff heart ischemia/reperfusion protocols. (C) Images of representative cardiac sections from C57/BL6 mouse and NMR post I/R. Red colour- viable tissue, white areas- post-infarct scar. (D) Reduced infarct size in NMR vs C57BL6 post-myocardial ischemia (20 min) and reperfusion (120 min) protocol. ¹H nuclear magnetic resonance spectroscopy analysis of cardiac metabolites post-20-minute ischemia in NMR vs C57/BL6 mouse, (E) elevated total adenine nucleotide pool (sum of ATP+ADP+AMP), (F) elevated myocardial glucose, (G) elevated myocardial lactate (H) reduced effluent lactate (I) reduced myocardial succinate and (J) reduced effluent succinate (1 min reperfusion). n=5/group, t-test, data normality tested by Shapiro-Wilk. *P<0.05, **P<0.01 *** P<0.001 **** P<0.0001.

MCT1 expression was analysed by western blotting. The blot in supplementary figure 1C looks like MCT1 is either the same or even lower in the naked mole-rat. The authors use Vinculin as housekeeping protein, but this protein from the blot shown seems to have a species specific expression as well, where it looks like vinculin is less expressed in naked mole-rat hearts. This issue was raised by me in the first round of revision and the authors have replied below that Tubulin seems to be a better housekeeping protein so perhaps Tubulin should be used to normalise MCT1. I understand Tubulin size is similar to MCT1, however the authors could strip and reprobe their MCT1 blot or run a parallel gel with the same samples. Caution has to be exercised when normalising to a housekeeping gene/protein to compare across species and choose a protein that is truly expressed equally in the two species. Finding such a protein is challenging. If such a protein is not found, it is best to avoid normalising and rather rely on whole protein staining via Ponceau or Coomassie. Assuming the authors attempt to load equal protein concentrations for their samples, and judging from the MCT1 gel alone, I cannot conclude a 3 fold higher MCT1 expression as quantified in figure 7B.

To conform this finding, the authors could perform a qPCR and a western blot with whole protein staining as control.

We acknowledge Reviewers critique and have removed MCT1 as well as LDH results from our manuscript (main manuscript figure as well as the supplementary figure S2). We have placed effluent lactate data into new Figure 6H (see previous figure).

3. To the new Figure 7

The sentence about MCT1 and LDH in mitochondria is highly speculative. This idea is still highly controversial and the authors have not provided a reference for this statement. The authors should back this up with a quantification of MCT1 and LDH from mitochondrial fractions.

We have removed MCT1 as well as LDH results from our manuscript (main manuscript figure as well as the supplementary figure S2).

In address of Major comments:

Point 4:

Unless more precise validation is done for detection of insulin with the mouse/rat kit, for example blood insulin after overnight fast versus refeeding versus bolus of glucose, the results from a mouse/rat based insulin kit may give false negatives. Alternatively the authors can use a guinea pig based ELISA kit as the naked mole-rat insulin has a similar mutation as all other hystricognaths including guinea pigs.

In order to address Reviewer's critique we have performed additional analysis using Guinea Pig Elisa [Guinea pig Insulin (INS) ELISA, Abbexa, USA, Catalogue number abx150418].

We have confirmed that naked mole rats have significantly lower insulin versus C57/BL6 mice. We have now amended our manuscript to reflect new guinea pig ELISA experiments (methods) and replaced previous ELISA results (new supplementary figure S4 B, shown below).

Point 7c: In interpreting the table with HIF1 a targets in different mole-rats. Is HIF1 alpha also stably expressed in other mole-rat species? This information would help to make sense of the Table supplied.

Hif1 mRNA level comparison between NMR and other mole rats is shown in our volcano plots (Figure 3). We have also added this information to the table (below) from our previous rebuttal (point 7c).

In normoxia, relative HIF1alpha protein expression was found to be comparable in the brains of *C. h. mahali*, *C. h. pretoriae*, *C. h. hottentotus*, *B. suillus*, *G. capensis* and *H. glaber* (PMID: 32041803).

Gene	Encoded protein	NMR vs Damaraland mole-rat	NMR vs Cape dune mole-rat	NMR vs Cape mole-rat	NMR vs Common mole-rat	NMR vs Highveld mole-rat	NMR vs Mahali mole-rat	NMR vs Natal mole-rat	NMR vs Golden mole
HIF1A	Hif1alpha	NS	NS	NS	down	down	down	down	down
ENO1	Enolase 1	down	NS	NS	NS	down	NS	NS	NS
ENO2	Enolase 2	NS	up	up	NS	down	down	down	down
LDHA	Lactate dehydrogenase A	down	down	down	down	down	down	down	down
PFKFB2	6-Phosphofructo-2-kinase/fructose-2,6-biphosphatase 2 (cardiac type)	up	up	up	up	up	up	up	up
PFKFB4	6-Phosphofructo-2-kinase/fructose-2,6-biphosphatase 4 (testes type)	NS	NS	up	up	up	up	up	up
SLC2A4	Glucose transporter 4	NS	NS	NS	NS	up	NA	up	up
VEGFA	Vascular endothelial growth factor A	NS	NS	NS	NS	NS	NS	NS	NS
VEGFB	Vascular endothelial growth factor B	NS	NS	down	NS	NS	NS	NS	NS
VEGFC	Vascular endothelial growth factor C	NS	down	down	down	down	down	down	down
VEGFD	Vascular endothelial growth factor D	down	down	NS	down	NS	NS	down	down

SLC16A4	Monocarboxylate transporter 5	NS	NS	up	up	NS	NS	NS	up
SLC16A7	Monocarboxylate transporter 5	up	up	up	up	up	up	up	up

Point 8:

Does running the experiment outside of naked mole-rat thermoneutral zone, ie. at temperatures much higher 37 rather than 32 affect heart function?

We haven't measured NMR function at both 37°C and 32°C in order to make the comparison. However, naked mole rat cardiac function at their thermoneutral zone vs C57/BL6 mouse (37°C) has been previously studied *in vivo* (PMID: 24363308).

We have also added the following study limitation section to our discussion paragraph (Page 9):

“In the current study, we have investigated differences between C57/BL6 mice, the golden mole and mole-rats including NMRs under uniform basal O₂ conditions (normoxia) and carried out cardiac perfusions at 37°C for C57/BL6 mice and NMRs. Thus, a potential limitation of our study is that it was performed in conditions not representative of the NMR environment.”

Point 10:

To support the minor damage post-ischaemia in the naked mole-rat heart which is a remarkable finding, do the authors have LVDP measurement for the reperfusion interval. What was the % recovery from the baseline measurement reported in Suppl Figure S5.

Unfortunately, we do not have this data available as we aimed to quantify infarct size following ischaemia and not to study functional recovery.

Further comment on Point 10:

I am surprised by how small the naked mole-rats hearts are compared to body size. In the paper by Grimes et al, (PMID: 24363308) reports a heart:body mass % 0.41 for naked mole-rat and 0.5 for mouse. The above publication is therefore in discrepancy of the results presented here. A 3 fold smaller heart proportionally is noteworthy and would have a significant impact on heart function. Can the authors comment on this. Furthermore in the Table S1 where the authors provide details for all African mole-rats, all the hearts from the mole-rat species seem strangely small. Is this a feature of the African mole-rat family? I was particularly surprised by the *suillus* mole-rat where the authors report a heart weight of 88.1mg for a 639g animal compared to for example *Cryptomys h. h.* with an 82.6 mg heart and weighing just 84 g. Could the authors check this data please and if this is really the case, this is quite astounding physiology given that generally heart:body weight is proportional across mammals (see figure 32.1 from <https://doi.org/10.1016/B978-0-12-385471-1.00032-5>)

We have updated the Supplementary Table 1. This has been due to error on our part and we have revisited all the morphology data (body weights, heart weights) which we have now addressed. Heart weight/body weight (mg/g) of NMR is 4.4 mg/g bw (=0.44%) vs mouse 6.2 mg/g body weight (0.62%), in line with published data.

Supplementary Table

Species	Common name	This study			Heart weight (mg)	Heart weight/body weight	Maximum lifespan (years)¹	Soil Type
		Sample size	Body weight (g)	Age (years)				
Heterocephalus glaber	Naked mole-rat	n=14 (7m, 7f)	32.9±4.8	5	178.0±33.6	4.4±0.6	>37	Hard clays
Georchys capensis	Cape mole-rat	n=8 (2m, 6f)	154±16.9	5	571.7±63.1	3.3±0.2	11	Predominantly soft clays but sometimes loams
Bathyergus suillus	Cape-dune mole-rat	n=7 (5m, 2f)	639±59.8	4-5	1642.5±219.0	2.8±0.3	>6	Loose coarse sands
Cryptomys hottentotus hottentotus	Common mole-rat	n=8 (6m, 2f)	84.2±6.9	4-5	343.9±51.7	4.0±0.3	11	Hard Clays and loams
Cryptomys hottentotus pretoriae	Highveld mole-rat	n=8 (3m, 5f)	102.7±11.1	4-5	559.8±50.9	4.7±0.3	11	Hard Clays and loams
Cryptomys hottentotus mahali	Mahali mole-rat	n=5 (1m, 4f)	114.8±8.6	4	N/A	N/A	11	Hard Clays and loams
Cryptomys hottentotus natalensis	Natal mole-rat	n=9 (3m, 6f)	82.3±6.7	4	469.9±13.2	4.3±0.04	11	Compacted thick Clays
Fukomys damarensis	Damaraland mole-rat	n=11 (3m, 8f)	124.6±8.2	6-7	583.3±32.6	4.3±0.3	16	Soft Coarse sands and compacted fine kalahari sands
Amblysomus hottentotus	Hottentot golden mole	n=3 (1m, 2f)	58.6±8.6	Adult (>1year)	N/A	N/A	>1	Compacted thick clays
Mus musculus	C57/BL6 mouse	n=5 (5m)	25.0±0.1	0.1	155.3±8.7	6.2±0.3	2-3	N/A

Point 11:

In Figure 4a, in the heatmap, ATP comes up as a separate metabolite. I am assuming for the ischaemic hearts, the authors used a similar method. Why can't the adenine pools be separated now?

We now provide the individual components of total adenine nucleotide pool post-ischemia as the new Supplementary Figure S7. Unfortunately, we are unable to distinguish ATP and ADP by ^1H nuclear magnetic resonance spectroscopy.

Fig S7. Individual components of myocardial total adenine nucleotide pool in post-ischemia.
Data mean + SEM, * $P < 0.05$ by t-test.

Furthermore thank you for pointing to the datasets, unfortunately the two links don't work and bring up a page saying DOI not found. Please check the link.

We have contacted Dryad and have asked for settings to be changed thus the Reviewer should have access to RNA sequencing data.

We have been advised to provide the Reviewer with the temporary sharing links as the dataset is in Private for Peer Review status. A substantial amount of space is required to download and unpack the gene sequencing files (not immediately apparent as the system is set to automatically start a download once you click on the link).

https://datadryad.org/stash/share/IBwk0EywLKRgKymb4pnj7mnbZqqPe6CWxRO2-_Ce8Jc

https://datadryad.org/stash/share/VGjaIJQatENOpJunpH9fFnkdKkIEIYrTsdYZf_NnQh8

Point 12:

Here I was more interested in the naked mole-rat heart and not the mouse heart which indeed has been reported several times. I am still not convinced whether the naked mole-rats do or do not accumulate succinate with ischaemia. The references provided by the authors emphasize the importance of looking at the same tissue set-up for succinate measurements, where one study looked within the same heart slice at injured versus healthy zone PMID: 35959683 or performed what I suggested above, succinate measurement after equilibration before ischaemia, and during/directly after ischaemia PMID: 32766828. In order to state that there is no succinate accumulation we need a baseline measurement for naked mole-rats, considering especially that the baseline was analysed from hearts dissected directly from animals connected to their circulation and at 28-32 degrees (thermoneutral zone of naked mole-rat) versus succinate measurement from ischaemic hearts maintained ex vivo and at 37 degrees (a stressful temperature for naked mole-rat physiology). These are two very different set ups to make direct comparisons regarding succinate levels. Furthermore proportionately the succinate levels between mouse and naked mole-rat in baseline and ischaemia do not vary much.

We respectfully disagree with the Reviewer. We do have a measurement of succinate under baseline normoxic conditions from the in vivo metabolomics which were not significant between C57/BL6 and NMR. Therefore, significantly lower succinate in NMR after ischaemia must reflect decreased succinate accumulation. To clarify this point we have now added succinate measured in the effluent during ischaemia which is also lower in NMR than C57/BL6 and therefore in agreement with this finding (Figure 6J).

Point 13:

This important point was not addressed by the authors.

Alternatively, if access to naked mole-rats may be a problem, but optimally in addition to the experiment in point 12, evidence of succinate accumulation in ischaemia should be further supported by looking at other TCA cycle intermediates. In another publications by some of the same authors of the current article (Chouchani 2014), citrate, aconitate, alpha-ketoglutarate, fumarate and malate have been shown to be unchanged under ischaemia relative to normoxic conditions whereas succinate accumulates 12-fold. Therefore, the authors should put in effort to show succinate in relation to the other TCA cycle intermediates and their changes relative to normoxic conditions to really pinpoint the lack of succinate accumulation and not just a reduced overall mitochondrial metabolism in the naked mole-rat.

Whilst we acknowledge that in the study we co-authored with Chouchani in 2014 we measured other TCA cycle intermediates in relation to succinate concentration post-ischemia, the aim of these two studies is different. Unlike 2014 Chouchani et al. paper, the aim of our present study was not to mechanistically study succinate metabolism in relation to RET. Our ischemia study was a functional follow up to gene and metabolomic comparison of NMR vs other mole rats as well as C57/BL6 mouse. Furthermore, TCA cycle intermediates are too low to be detected by ¹H NMR spectroscopy and we have no tissue available to seek additional means of

quantification. We have however now included succinate measured in the effluent to support our finding that succinate accumulation is lower in NMR during ischaemia.

Point 14:

From the data presented the only conclusion that can be reached is that there is generally less succinate in the naked mole-rat, in both normal and ischaemic conditions. This may be because of lower/less efficient mitochondria, lowered heart rate/metabolic rate, or some other parameters. I am not sure how the succinate measurement in their set-up relates to ischaemia as we do not have a controlled baseline succinate measurement to compare succinate under ischaemia with. Please see my comment for Point 12.

We respectfully disagree with this point as we measured succinate in two separate studies (including this one) in non-ischemic NMR hearts and found comparable levels of succinate.

Point 15:

MCT1 and LDH at mitochondria is highly speculative and should be supported by blots from mitochondrial fractions or omitted.

We agree with the Reviewer and have removed all LDH and MCT1 data as well as discussion from our manuscript.

Point 17:

Such analyses have been performed previously for other tissues like liver and could help to support some of the authors findings for the rest of the manuscript which focussed mainly on comparing mouse and naked mole-rat. Have the authors tried to consult previous literature of cross-species RNAseq DEGs analysis and tried to apply similar methods?

See DOI:10.1126/science.aau0236, doi: 10.1186/s12915-018-0546-z

We have performed additional mouse versus NMR DEG analysis as requested by the Reviewer. This is a workaround due to the standard analysis pipeline at BGI (RNA sequencing commercial service provider) failing to hit the mapping criteria for mouse versus NMR genome and vice versa, as we stated in the manuscript.

It is important to note that this approach (described below and now added to the methods) can only capture genes that are expressed in *both species*, so of the top 20 expressed genes in the NMR, only FOUR are also expressed in the mouse and NONE of the top 12 in the NMR are also expressed in the mouse, (as we have already reported). Thus, we cannot calculate a log fold change (LFC) for these important genes as the mouse has zero counts. Of note, there can be a high log fold change for genes that are way down on the expression/read counts list and therefore may be of limited significance.

The workaround approach we took to attempt a DEG analysis between the mouse and NMR was to take the list of NMR genes identified by BGI and download them from ENSEMBL to our bioinformatics cluster. Then, we downloaded the mouse genome and BLASTING the former against the latter one at a time (with an 80% similarity threshold and strict e-value threshold of 1e-50) we were able to match the NMR genes with the mouse gene identifier, find the corresponding transcript counts and work out log fold changes (LFC). We were quite successful with this approach, of 18,012 NMR genes we got corresponding mouse genes for 12,206. Of these we were able to calculate log fold changes for 7,949 genes that were expressed in both the mouse and the NMR (see new Supplementary Data Table S4).

We now include a Manhattan plot (Supplementary Figure S1) with these genes arranged in alphabetical order on the x-axis and LFC on the y-axis, highlighting the top 1% in a different plot colour and labelling the top 10 genes by name (blue for the NMR and red for the mouse; new Supplementary Figure S1). We have also amended our manuscript to include new method and results (green font).

Results, page 5.

“We have also performed additional NMR versus C57/BL6 mouse DEG analysis (Online Supplement Fig S1). Of the 18,012 NMR genes assessed we could analyse 12,206 corresponding mouse genes. Out of these we were able to calculate log fold changes for 7,949 genes that were expressed in both the mouse and the NMR (Supplementary Data Table S4). Of the top 20 expressed genes in the NMR, only four are also expressed in the mouse and none of the top 12 in the NMR are also expressed in the mouse. The gene *Clgn* has the highest log fold change in NMRs versus the mouse, but it is ranked 1760 in genes ordered by expression (FPKM) in the NMR.”

Online Supplement Results

Fig. S1. Manhattan plot Log fold change gene expression NMR vs C57/BL6 Manhattan plot with top 1% gene expressed highlighted and top 10 genes labelled by name (blue for the NMR and red for the mouse). Note that of the top 20 expressed genes in the NMR, only FOUR are also expressed in the mouse and NONE of the top 12 in the NMR are also expressed in the mouse. The gene “clgn” has the highest logfold change but it is rank 1760 in genes ordered by expression (FPKM) in the NMR.

Methods, Page 11

“Due to limited mapping rate using the naked mole-rat as a reference for the comparison with the mouse C57/BL6 transcriptome (and *vice versa*), and the limited common expression of the most expressed genes in each species, it was not possible to do the same range of analyses for the mouse versus the naked mole-rat (or mouse versus the other species), as for the naked mole-rat versus the subterranean species listed above. Instead, as a workaround, we identified orthologous genes in the naked mole-rat and mouse in order to compare gene expression between the species. For transcripts that mapped to the naked mole-rat genome, we extracted the corresponding gene sequences directly from the naked mole-rat genome (GCA_944319725.1). For novel transcripts, we used associated protein IDs identified by BGI and downloaded the corresponding gene sequences using the Entrez Direct Utilities³⁵. We used the Basic Local Alignment Search Tool (BLAST; v2.11.0³⁶) with parameters `-evalue 1e-50 -perc_identity 80` to query all sequences against the mouse reference genome (GCF_000001635.27). We took the best hit for each transcript and calculated the log₂ fold change between the mean abundance (FPKM) of each species. We plotted the data using the R package ggplot2.^{35,36}”

Reviewer #4 (Remarks to the Author):

The authors have addressed all of my comments appropriately and I have no further changes to suggest.

No response required.

REVIEWERS' COMMENTS

Reviewer #1 (Remarks to the Author):

This paper provided valuable resources for comparative biologists and has addressed and corrected all of my prior concerns.

Reviewer #2 (Remarks to the Author):

The author revisions fully deal with issues raised by me, and matters I also considered regarding Reviewer 1 as referred to me by the editor.

Reviewer #3 (Remarks to the Author):

I thank the authors for revising the manuscript and correcting several critical errors in their publication. One outstanding issue I have is with Point 12:
Could the authors check the statistical test and result used to get non-significance for succinate between naked mole-rat and mouse at baseline (Fig4F). When compared to the statistics of creatine (Fig4E), it would seem to me that succinate between naked mole-rats and mice is more significantly different than creatine, yet for creatine the p-value reported is >0.01 and for succinate it is "ns". This would impact the statement the authors are making regarding the reduced accumulation of succinate in ischemia in point 12, 13, and 14.

Reviewers Response

Reviewer #1 (Remarks to the Author):

This paper provided valuable resources for comparative biologists and has addressed and corrected all of my prior concerns.

No response required.

Reviewer #2 (Remarks to the Author):

The author revisions fully deal with issues raised by me, and matters I also considered regarding Reviewer 1 as referred to me by the editor.

No response required.

Reviewer #3 (Remarks to the Author):

I thank the authors for revising the manuscript and correcting several critical errors in their publication. One outstanding issue I have is with Point 12:

Could the authors check the statistical test and result used to get non-significance for succinate between naked mole-rat and mouse at baseline (Fig4F). When compared to the statistics of creatine (Fig4E), it would seem to me that succinate between naked mole-rats and mice is more significantly different than creatine, yet for creatine the p-value reported is >0.01 and for succinate it is "ns". This would impact the statement the authors are making regarding the reduced accumulation of succinate in ischemia in point 12, 13, and 14.

We thank the Reviewer for their point. We confirm that we have reviewed the data and checked the statistical analysis and it is correct, the result is non-significant: there is no difference in myocardial succinate levels in normoxia between naked mole rats and other mole rats.

However, on reviewing the data we noted a small error in our previous presentation which we have now corrected. The data has an outlier detection applied prior to the PCA and heatmap analysis in Figures 4A and B. Unfortunately this got transferred to the univariate analysis in Figures 4C-F. We have replotted the raw data prior to the outlier detection in Figures 4C-F and have added the pairwise p values for all comparisons. Details of the statistical test used is displayed in the updated Figure 4 legend. This was a one-way ANOVA of each pairwise comparison with Dunnett's multiple comparisons test. This does not affect the result, nor does it alter our previous interpretation.

Figure 4. Enhanced cardiac energy reserve is unique to naked mole rat. (A) Heatmap summary of the cardiac metabolite concentrations determined by high resolution ^1H nuclear magnetic resonance spectroscopy showing distinct profile of naked mole rat (NMR, $n = 6$) versus Cape mole-rat ($n = 8$), Natal mole-rat ($n = 9$), Common mole-rat ($n = 8$), Damaraland mole-rat ($n = 11$), Highveld mole-rat ($n = 8$), Cape dune mole-rat ($n = 7$), Mahali mole-rat ($n = 5$), Hottentot golden mole ($n = 3$) and C57/BL6 mouse ($n = 5$), where n correspond to biological repeats. The colorbar denotes the number of standard deviations above or below the row mean of each metabolite. Dendrograms top and left side show hierarchical cluster analysis of related genotypes (columns) and related metabolites (rows). Data are standardized along each row (metabolite) so the mean is 0 and the standard deviation is 1. (B) Linear discriminant analysis of the metabolite data in panel A showing clear separation of NMR from all other genera. Ellipses correspond to 95% confidence intervals. Unsupervised Principal Component Analysis (PCA) plot is shown in Supplemental Fig S2. (C) NMR is characterised by supra-normal myocardial glycogen content, (D) elevated glucose-1-phosphate, the product of glycogen catabolism, (E) elevated cellular energy reserve creatine, while (F) succinate was not significantly different under baseline normoxic conditions. P values calculated from a one-way ANOVA of each pairwise comparison with Dunnett's multiple comparisons test. Data are presented as mean \pm SEM, * $p < 0.05$, ** $p < 0.01$ *** $p < 0.001$ **** $p < 0.0001$. Source data are provided as a Source Data file.